# ParisKV: Fast and Drift-Robust KV-Cache Retrieval for Long-Context LLMs

Yanlin Qi [1]  Xinhang Chen [2]  Huiqiang Jiang [3]  Qitong Wang [4]  Botao Peng [5]  Themis Palpanas [1]

## Abstract

KV-cache retrieval is essential for long-context LLM inference, yet existing methods struggle with distribution drift and high latency at scale. We introduce ParisKV, a drift-robust, GPU-native KV-cache retrieval framework based on collision-based candidate selection, followed by a quantized inner-product reranking estimator. For million-token contexts, ParisKV supports CPU-offloaded KV caches via Unified Virtual Addressing (UVA), enabling on-demand top-$k$ fetching with minimal overhead. ParisKV matches or outperforms full attention quality on long-input and long-generation benchmarks. It achieves state-of-the-art long-context decoding efficiency: it matches or exceeds full attention speed even at batch size 1 for long contexts, delivers up to $2.8\times$ higher throughput within full attention's runnable range, and scales to million-token contexts where full attention runs out of memory. At million-token scale, ParisKV reduces decode latency by $17\times$ and $44\times$ compared to MagicPIG and PQCache, respectively, two state-of-the-art KV-cache Top-$k$ retrieval baselines; code is available at https://github.com/amy-77/ParisKV/tree/main.

## 1. Introduction

Large language models (LLMs) (Achiam et al., 2023; Touvron et al., 2023; Jiang et al., 2023; Yang et al., 2025) are rapidly extending their context windows to millions of tokens, placing unprecedented pressure on inference efficiency. The KV-cache, which stores keys and values of all preceding tokens for autoregressive decoding, quickly dominates both memory footprint and latency as context

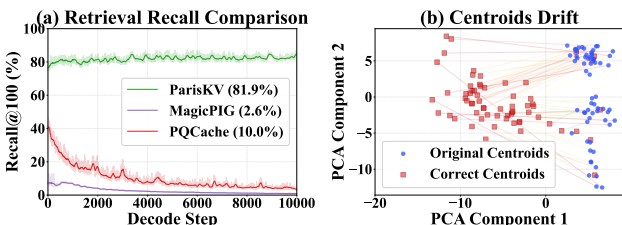

Figure 1. Retrieval drift results. (a) Recall comparison of different methods on AIME. (b) Centroid drift induced by decoding keys, measured as the mismatch between *prefill-only* centroids (original centroids in blue) and *reference* centroids (correct centroids in red) obtained by clustering all keys from both prefill and decoding.

grows (Zhang et al., 2023; Li et al., 2024; Chen et al., 2024). Crucially, long-context decoding is memory-bound: each step requires reading a large volume of KV vectors, and bandwidth cost scales linearly with context length. This has motivated sparse/selective attention, which exploits the empirical sparsity of attention by attending only to a Top-$k$ subset of relevant past tokens.

Among sparse designs, KV-cache dropping/eviction methods permanently discard tokens and can be brittle when early tokens become critical later. In contrast, KV-cache retrieval retains the full KV history and dynamically retrieves relevant keys at each step, making it better suited for open-ended long-context inference.

**Challenges**: Despite its accuracy advantages, KV-cache retrieval faces the following efficiency challenges in latency-sensitive, long-generation scenarios. (C1) Speed–quality tradeoff: Lightweight approximate retrieval (e.g., coarse clustering, low-bit quantization) sacrifices recall for speed; recovering accuracy requires increasing the retrieval budget, eroding the benefits of sparsity. (C2) Decoding drift: Centroids learned from clustering prefill keys become stale as generation continues. Figure 1(a) shows an example, where prior methods (Zhang et al., 2025b; Chen et al., 2024) suffer severe recall collapse during long decoding, while ParisKV maintains stable recall. Figure 1(b) visualizes the root cause: prefill-only centroids (in blue) increasingly mismatch the true key distribution (in red) as decode keys accumulate. (C3) CPU-bound retrieval and data movement: When the full KV cache is offloaded to CPU memory, retrieval typically involves CPU-side search and CPU-to-GPU data transfer. The GPU can only access representations (centroids or low-bit codes), which introduce approximation errors, while

[1]LIPADE, Université Paris Cité, Paris, France [2]Xi'an Jiaotong University, Xi'an, China [3]Qwen Team, Alibaba Group, China [4]Harvard University, Cambridge, MA, USA [5]Institute of Computing Technology, Chinese Academy of Sciences, Beijing, China. Correspondence to: Botao Peng <pengbotao@ict.ac.cn>.

*Proceedings of the 43rd International Conference on Machine Learning*, Seoul, South Korea. PMLR 306, 2026. Copyright 2026 by the author(s).

CPU orchestration dominates end-to-end latency.

**Our approach.** We present ParisKV, a hardware-aware algorithm–system co-design for KV-cache retrieval that achieves fast decoding, stable recall under long input and/or long-generation (i.e., million-token scalability), even in the presence of drift.

Unlike prior methods that learn centroids from prefill keys, ParisKV transforms queries and keys into a stable, data-independent space by normalizing them onto the unit hypersphere (see Fig. 3) and applying a shared random orthogonal rotation, where we can analytically define a set of uniformly distributed centroids. These centroids are uniformly distributed on the unit hypersphere, so any newly generated keys will always be close to at least one centroid. In this way, ParisKV avoids stale centroids (i.e., centroids that become mismatched during decoding), and maintains stable retrieval quality throughout long generations.

ParisKV adopts a two-stage pipeline implemented entirely on GPU. The coarse stage uses multi-subspace collision counting to aggressively prune candidates; the fine stage reranks them using calibrated inner-product estimates from compact representation of original keys, achieving high recall without accessing full-precision keys on CPU.

Moreover, ParisKV offloads the full KV cache to CPU memory and leverages Unified Virtual Addressing (UVA) to let GPU kernels directly fetch only the selected Top-$k$ KV pairs on demand, bypassing explicit memcpy and CPU-side scheduling overhead.

**Contributions. (1)** We propose drift-robust analytic centroids, a data-independent transformation that maintains stable retrieval recall even under significant distribution shift at the million-token scale. **(2)** We design a GPU-native coarse-to-fine retrieval pipeline combining collision-based pruning and calibrated reranking, achieving high-fidelity Top-$k$ selection directly on compressed codes without CPU-side search. **(3)** We implement a scalable UVA-based offloading system enabling on-demand KV fetching with minimal CPU intervention. **(4)** Extensive evaluations show ParisKV delivers up to $17\times$ and $45\times$ speedups over MagicPIG and PQCache at 1M-token scale, while maintaining near-lossless quality on long-input and long-form reasoning benchmarks.

**Conflict of Interest Disclosure.** The authors declare no financial conflicts of interest.

## 2. Related Work

### 2.1. KV-Cache Quantization

Several recent studies have proposed innovative methods to tackle the challenges of KV-Cache quantization while keeping performance degradation minimal.

Quantization-friendly transformations reduce quantization difficulty via offline distribution reshaping. SmoothQuant (Xiao et al., 2023a) enables 8-bit KV-Cache quantization, while QuaRot, QuQuant, and SpinQuant further suppress outliers through randomized or learnable transformations (Ashkboos et al., 2024; Lin et al., 2024; Liu et al., 2024c). Sparsity-aware quantization combines importance modeling with low-bit quantization; Q-Hitter (Zhang et al., 2024) leverages attention statistics to jointly apply sparsity and 8-bit quantization. Structure-aware and mixed-precision quantization exploits KV heterogeneity via offline tuning: KIVI (Liu et al., 2024b) applies asymmetric quantization for 2-bit compression, while KVTuner (Li et al., 2025b) determines layer-wise mixed-precision configurations.

### 2.2. Sparse KV-Cache

Sparse KV-Cache methods reduce the memory and computation cost of long-context inference by exploiting attention sparsity. Existing work can be broadly categorized into *KV-Cache Dropping* and *KV-Cache Retrieval*.

**KV-Cache Dropping.** KV-Cache Dropping permanently removes KV entries that are predicted to be unimportant. StreamingLLM (Xiao et al., 2023b) treats early tokens as sinks, while H2O (Zhang et al., 2023), SnapKV (Li et al., 2024), Keyformer (Adnan et al., 2024), and Expected Attention (Devoto et al., 2025) estimate token importance from attention statistics. KeyDiff (Park et al., 2025) infers importance from key diversity, enabling compatibility with optimized attention implementations (Dao et al., 2022; Dao, 2023). Structure-aware methods further exploit head and layer-level heterogeneity, including DuoAttention (Xiao et al., 2024), PyramidKV (Cai et al., 2024), AdaKV (Feng et al., 2024), and HeadKV (Fu et al., 2024). KVZip (Kim et al., 2025) identifies important historical KV entries via offline reconstruction.

**KV-Cache Retrieval.** KV-Cache Retrieval dynamically selects a small subset of KV entries per step without permanently removing them. Pattern-aware sparse attention mainly targets the prefill stage, including MInference (Jiang et al., 2024), FlexPrefill (Lai et al., 2025), XAttention (Xu et al., 2025), and SeerAttention (Gao et al., 2024). Decode-stage retrieval is explored by Quest (Tang et al., 2024), while MoBA (Lu et al., 2025) and NSA (Yuan et al., 2025) integrate retrieval mechanisms into training. CPU-assisted retrieval further extends effective KV capacity, such as MagicPig (Chen et al., 2024), PQCache (Zhang et al., 2025b), RetroInfer (Chen et al., 2025), ShadowKV (Sun et al., 2024), and RetrievalAttention (Liu et al., 2024a). External-knowledge retrieval, including graph-based RAG frameworks (Zhang et al., 2025a), is complementary to our setting: ParisKV retrieves from the model's internal KV

cache rather than from an external corpus.

## 3. Background and Problem Formulation

We study *KV-cache retrieval* for accelerating autoregressive decoding in Transformers. At decoding step $t$, each attention head issues a query vector $\mathbf{q} \in \mathbb{R}^D$, and attends to a growing set of keys $\mathcal{K} = \{\mathbf{k}_i \in \mathbb{R}^D\}_{i=1}^{n_t}$ with corresponding values $\mathcal{V} = \{\mathbf{v}_i\}_{i=1}^{n_t}$ stored in the KV cache. The pre-softmax attention score between a key and the query is $s(\mathbf{k}_i, \mathbf{q}) \triangleq \langle \mathbf{k}_i, \mathbf{q} \rangle$. In exact attention, we compute a softmax over *all* cached keys and aggregate values:

$$\alpha_i(\mathbf{q}) = \frac{\exp\left(s(\mathbf{k}_i, \mathbf{q})\right)}{\sum_{j=1}^{n_t} \exp\left(s(\mathbf{k}_j, \mathbf{q})\right)}, \qquad \mathbf{o}(\mathbf{q}) = \sum_{i=1}^{n_t} \alpha_i(\mathbf{q})\,\mathbf{v}_i. \tag{1}$$

However, the KV cache grows with $t$, making Eq. (1) increasingly expensive.

**Top-$k$ retrieval for approximate attention.** To reduce latency, we aim to retrieve only the top-$k$ keys with the largest inner products: $\mathrm{TopK}(\mathbf{q}) \triangleq \arg\max_{i\in[n_t]}^k \langle \mathbf{k}_i, \mathbf{q} \rangle$. Given a small candidate set $\mathcal{C}(\mathbf{q}) \subseteq [n_t]$ (ideally containing $\mathrm{TopK}(\mathbf{q})$), we compute *approximate attention* by restricting the softmax to $\mathcal{C}(\mathbf{q})$:

$$\tilde{\alpha}_i(\mathbf{q}) = \frac{\exp\left(s(\mathbf{k}_i, \mathbf{q})\right)}{\sum_{j\in\mathcal{C}(\mathbf{q})} \exp\left(s(\mathbf{k}_j, \mathbf{q})\right)}, \quad i \in \mathcal{C}(\mathbf{q}). \tag{2}$$

$$\tilde{\mathbf{o}}(\mathbf{q}) = \sum_{i\in\mathcal{C}(\mathbf{q})} \tilde{\alpha}_i(\mathbf{q})\,\mathbf{v}_i. \tag{3}$$

When $\mathcal{C}(\mathbf{q})$ covers the true top-$k$ keys, Eq. (3) closely matches the exact output in Eq. (1) while being faster.

## 4. The ParisKV Framework

ParisKV is an algorithm–system co-design for fast long-context decoding with *drift-robust* retrieval. We first provide an overview of ParisKV, and then discuss the details.

In the *prefill* phase (refer to Fig. 2), beyond computing the KV cache, ParisKV performs a one-time *key summarization* to enable fast on-device retrieval during decoding. Specifically, we (i) apply a shared normalization-and-rotation transform to make representations stable under decoding drift, and (ii) build compact per-key metadata that stays on GPU after the full-precision KV cache is offloaded to CPU. These metadata have two parts. First, we use a small *codebook*, i.e., a small set of centroid IDs that summarize keys (A.2 in Fig. 2), which is later used to quickly generate candidates via collision-based voting (B.2.1 in Fig. 2). Second, we store a compact quantized code per subspace for each key (A.3.1), together with a lightweight scaling factor for calibration (A.3.2). These quantized codes and scaling factors are then used to efficiently estimate query–key inner products and

rerank candidates (B.2.2) without accessing full-precision keys on GPU. Overall, prefill produces small, GPU-resident summaries for retrieval, while the full-precision KV cache is asynchronously offloaded to CPU memory to allow scalability to long contexts.

In the *decode* phase (refer to Fig. 2 and Fig. 4), ParisKV uses a two-step retrieval pipeline. (1) Coarse Candidate Generation (B.2.1) narrows the search space using only lightweight GPU-resident *codebook* summaries. For a query, we select the few most similar centroids in each subspace; keys in these selected buckets receive one vote. We sum votes across subspaces and keep the high-vote keys as candidates. (2) Candidate Reranking reranks the candidates generated earlier by estimating their query-key inner products and selecting the final Top-$k$ keys for attention. During this step, the GPU maintains compact 4-bit quantized codes of the keys, but cannot (efficiently) access the full-precision raw vectors, which are in the CPU memory. Thus, ParisKV computes a fast, subspace-wise inner-product estimate using the quantized codes and corrects the quantization error using a lightweight procedure (see Fig. 4 bottom), achieving high fidelity, and consequently enabling high-recall in the Top-$k$ selection. Estimating the query–key inner products involves many fine-grained operations, so we implement a *reranking fused kernel* to streamline the computation.

### 4.1. Prefill Phase

#### 4.1.1. PREPROCESSING

ParisKV applies a shared normalize–rotate transform, splits vectors into rotated subspaces, and uses a polar (radius–direction) form to summarize each subspace (A.1 in Fig. 2). Given prefill keys $\{\mathbf{k}_i\}_{i=1}^n$, ParisKV constructs compact per-key metadata on GPU (queries are processed in the same way). It proceeds as follows.

**(1) Normalize & rotate.** Normalization maps all keys and queries onto the same unit-hypersphere space, so similarity comparisons depend only on direction (not magnitude). The shared rotation spreads information more evenly across dimensions while preserving inner products, making later bucketing and scoring more stable under drift. We first $\ell_2$-normalize keys and queries: $\hat{\mathbf{k}}_i = \frac{\mathbf{k}_i}{\|\mathbf{k}_i\|_2}$, $\hat{\mathbf{q}} = \frac{\mathbf{q}}{\|\mathbf{q}\|_2}$.

We then apply a shared orthogonal rotation $\mathbf{R}$ (implemented by SRHT): $\tilde{\mathbf{k}}_i = \mathbf{R}\hat{\mathbf{k}}_i$, $\tilde{\mathbf{q}} = \mathbf{R}\hat{\mathbf{q}}$. Since $\mathbf{R}$ is orthogonal, it preserves inner products: $\langle \hat{\mathbf{k}}_i, \hat{\mathbf{q}} \rangle = \langle \tilde{\mathbf{k}}_i, \tilde{\mathbf{q}} \rangle$.

*Example.* Assume $m = 3$. Then, the codebook has $2^3{=}8$ direction centers: $\Omega = \left\{ \left( \pm\frac{1}{\sqrt{3}}, \pm\frac{1}{\sqrt{3}}, \pm\frac{1}{\sqrt{3}} \right) \right\}, |\Omega| = 8$. These 8 centers correspond to the 8 corners of a cube. After scaling to unit norm, they lie on the unit sphere and evenly cover the 8 "octants" (i.e., eight coarse directions in 3D space). Given a rotated key $\tilde{k}$, we simply assign it to the nearest center

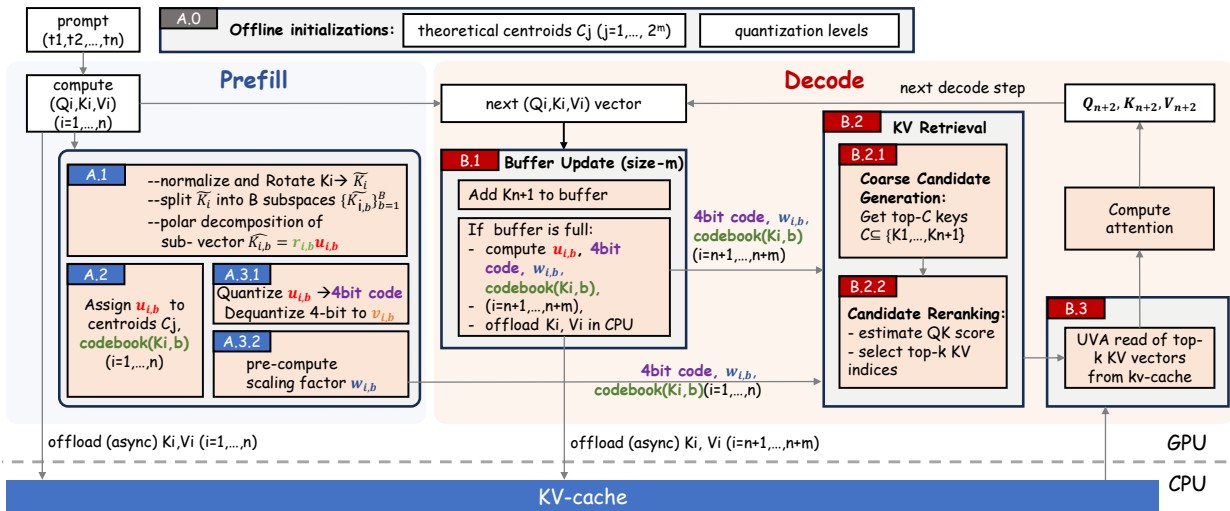

*Figure 2.* ParisKV pipeline. Offline, we construct an analytic centroid codebook and a quantization configuration. During prefill, we materialize the KV cache and build GPU-resident key summaries (centroid IDs for Stage-I vote-based filtering, and low-bit codes with lightweight weights for Stage-II reranking), while asynchronously offloading full-precision KV to CPU memory. During decoding, summaries for newly generated keys are incrementally updated; the GPU performs coarse-to-fine retrieval (voting → reranking) using only these summaries, then fetches the selected Top-$k$ KV pairs from CPU via UVA for attention.

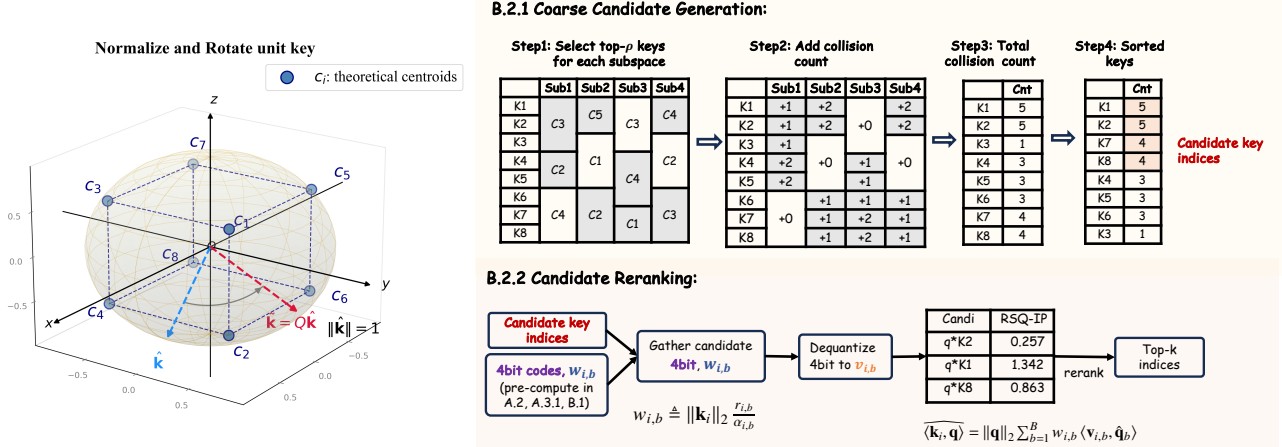

*Figure 3.* Illustration of rotation-based codebook assignment on the unit sphere.

*Figure 4.* ParisKV Retrieval algorithm (shown as B.2 in Fig. 2)

in $\Omega$. For instance, if $\tilde{k}$ points roughly to the "$(+,+,-)$" direction, it will be assigned to $\left(\frac{1}{\sqrt{3}}, \frac{1}{\sqrt{3}}, \frac{-1}{\sqrt{3}}\right)$. (The details are shown in Fig.3)

**(2) Subspace split.** Subspace splitting enables efficient parallel processing and provides multiple independent votes for our collision-based candidate filtering. We partition $\tilde{\mathbf{k}}_i$ (and $\tilde{\mathbf{q}}$) into $B$ contiguous subspaces of dimension $m = D/B$: $\tilde{\mathbf{k}}_i = [\tilde{\mathbf{k}}_{i,1}; \ldots; \tilde{\mathbf{k}}_{i,B}]$ and $\tilde{\mathbf{q}} = [\tilde{\mathbf{q}}_1; \ldots; \tilde{\mathbf{q}}_B]$, where $\tilde{\mathbf{k}}_{i,b}, \tilde{\mathbf{q}}_b \in \mathbb{R}^m$.

**(3) Polar decomposition.** Polar form further separates direction (for voting) from radius (for calibrated inner-product estimation). For each key $i$ and subspace $b$, define $r_{i,b} = \|\tilde{\mathbf{k}}_{i,b}\|_2$ and $\mathbf{u}_{i,b} = \tilde{\mathbf{k}}_{i,b}/r_{i,b}$, so that $\tilde{\mathbf{k}}_{i,b} = r_{i,b}\mathbf{u}_{i,b}$ (similarly for $\tilde{\mathbf{q}}_b$). This yields an additive subspace form of

the rotated inner product:

$$\langle \tilde{\mathbf{k}}_i, \tilde{\mathbf{q}} \rangle = \sum_{b=1}^{B} \langle \tilde{\mathbf{k}}_{i,b}, \tilde{\mathbf{q}}_b \rangle = \sum_{b=1}^{B} r_{i,b} \langle \mathbf{u}_{i,b}, \tilde{\mathbf{q}}_b \rangle. \qquad (4)$$

### 4.1.2. ASSIGN DIRECTIONS TO CENTROIDS

We map each $\mathbf{u}_{i,b}$ to its nearest direction centroid $\mathbf{c}_j$ (A.2 in Fig. 2), and store the centroid ids codebook to use during decoding for the collision-based candidate generation.

**Data-independent uniform direction centroids.** After $\ell_2$ normalization and SRHT rotation, subspace directions become approximately isotropic, making it effective to use a *shared, data-independent* set of direction centroids instead of learning them from data. Thus, in each $m$-dimensional

subspace we predefine a sign-pattern centroid set

$$\Omega \triangleq \left\{ \pm \frac{1}{\sqrt{m}} \right\}^m, \qquad |\Omega| = 2^m, \qquad (5)$$

where each $\boldsymbol{\omega} \in \Omega$ is a unit direction with equal-magnitude coordinates. Given a subspace unit direction $\mathbf{u}_{i,b}$, we assign it to the nearest centroid in $\Omega$ and store the resulting index

$$\text{centroid\_id}_{i,b} \triangleq \arg \max_{\boldsymbol{\omega} \in \Omega} \langle \mathbf{u}_{i,b}, \boldsymbol{\omega} \rangle. \qquad (6)$$

### 4.1.3. A.3 LOW-BIT RERANK METADATA

To rerank candidates without accessing full-precision keys in CPU memory, we store lightweight per-subspace summaries on-GPU: (A.3.1) a 4-bit direction code $\text{code}_{i,b}$ that dequantizes to a reconstructed direction $\mathbf{v}_{i,b}$, and (A.3.2) a scalar weight $w_{i,b}$ that precomputes all key-only factors during candidate reranking. Specifically, with subspace radius $r_{i,b} = \|\tilde{\mathbf{k}}_{i,b}\|_2$ and per-subspace correction term

$$\alpha_{i,b} \triangleq \langle \mathbf{v}_{i,b}, \mathbf{u}_{i,b} \rangle. \qquad (7)$$

where $\mathbf{u}_{k,b}$ is the (unknown) true unit direction of key $k$ in subspace $b$ and $\mathbf{v}_{k,b}$ is its quantized direction. Intuitively, $\alpha_{k,b}$ measures the alignment between the quantized direction and the true direction; quantization typically shrinks this alignment, causing a systematic underestimation of the query–key inner product if we directly use $\langle \mathbf{v}_{k,b}, \tilde{\mathbf{q}}_b \rangle$. Specifically, our reranking estimator employs the approximation

$$\langle \mathbf{u}_{k,b}, \tilde{\mathbf{q}}_b \rangle \approx \frac{\langle \mathbf{v}_{k,b}, \tilde{\mathbf{q}}_b \rangle}{\alpha_{k,b}}, \qquad (8)$$

we define a per-key, per-subspace scaling factor

$$w_{i,b} \triangleq \|\mathbf{k}_i\|_2 \frac{r_{i,b}}{\alpha_{i,b}}. \qquad (9)$$

Notably, $w_{i,b}$ depends only on the key and the quantization metadata, and does *not* involve the query. Therefore, it can be precomputed once during the prefill stage and cached, so that online reranking only computes lightweight dot products between the quantized direction and the rotated query, and then rescales them using the cached $w_{i,b}$ to estimate $\langle \mathbf{k}_i, \mathbf{q} \rangle$ efficiently. Online reranking then becomes a simple weighted accumulation over subspaces:

$$\widehat{\langle \mathbf{k}_i, \mathbf{q} \rangle} = \|\mathbf{q}\|_2 \sum_{b=1}^{B} w_{i,b} \langle \mathbf{v}_{i,b}, \tilde{\mathbf{q}}_b \rangle. \qquad (10)$$

The reranking inner-product estimation is detailed in Appendix B.2.2. Overall, prefill stores metadata $\{(\text{centroid\_id}_{i,b}, \text{code}_{i,b}, w_{i,b})\}_{i \leq n, b \leq B}$ on GPU, while full-precision $(\mathbf{k}_i, \mathbf{v}_i)$ are kept in CPU memory for on-demand UVA fetching.

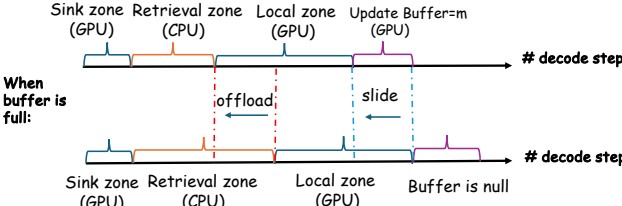

*Figure 5.* Sliding-window KV-cache update.

### 4.2. Decode Phase

#### 4.2.1. BUFFER UPDATE

As decoding proceeds, we maintain the KV-cache as four contiguous regions (Fig. 5): *Sink* (a small set of early high-attention tokens kept on GPU), *Retrieval* (offloaded and indexed historical tokens), *Local* (the most recent local_size tokens kept on GPU for dense attention), and *Update Buffer* on GPU that temporarily holds newly generated tokens.

We append each new token to the Update Buffer (B.1 in Fig. 2). When the buffer reaches m tokens, we trigger a sliding-window update: (i) the oldest m tokens in the Local region are evicted into the Retrieval region (via GPU→CPU transfer in the offload setting); (ii) the Local window is shifted to keep the most recent local_size tokens by promoting the buffered tokens into Local (and clearing the buffer); and (iii) we encode and index the evicted keys on GPU (centroid ids, 4-bit codes, and $w_{i,b}$), while offloading the corresponding full-precision KV pairs asynchronously. This streaming update supports unbounded generation while keeping retrieval metadata fresh and preserving fast access to both recent context (Local) and long-range history (Retrieval).

#### 4.2.2. RETRIEVAL

At each decoding step, ParisKV performs a fully on-GPU two-step retrieval pipeline (Fig. 4 and B.2 in Fig. 2) to approximate Top-$k$ attention selection without accessing full-precision keys during ranking.

**(1) Coarse candidate generation** (B.2.1). We use a direction-only proxy to quickly prune the full key set to a small candidate pool. Concretely, we decompose the rotated key/query into $B$ subspaces and score each key by accumulating subspace-wise collision bonuses: in each subspace we compare the query to the key's assigned centroid (cheap dot product $q^\top c$) and only let the top-$\rho$ fraction contribute a non-zero bonus. Summing across subspaces yields a coarse integer score that is robust to decoding drift and enables fast pruning to the top-$\beta$ fraction as candidates (typically $\beta$=5%–10%). We set $\rho \geq \beta$ so each subspace contributes sufficiently many non-zero terms, and the candidate pool remains comfortably larger than the final Top-$k$ (e.g., $k$=100) used in reranking. The details are shown in Appendix B.2.1.

**(2) Candidate Reranking** (B.2.2). We rerank candidates by estimating the raw attention score $\langle \mathbf{k}, \mathbf{q} \rangle$ directly from compact 4-bit key quantizations. Each subspace direction is encoded with *1-bit sign + 3-bit magnitude* per coordinate. Online reranking only computes lightweight dot products between the quantized direction and the rotated query, combined with a precomputed per-subspace scaling factor. This yields an accurate estimator of raw inner products while avoiding any access to full-precision keys during reranking. Full-precision KV are fetched *only* for the final selected Top-$k$ tokens. Appendix B.2.2 describes in detail the inner-product estimator used for reranking (including the correct term in Eq. (7) and approximation in Eq. (19)). The derivation of the theoretical quantization levels for the normalized-and-rotated subspace directions is guided by Proposition 4.1 (with full derivations in Appendix B.1.1).

**Proposition 4.1** (Rotation-induced Beta priors for subspaces). *Let $\hat{\mathbf{k}} \in \mathbb{S}^{D-1}$ be any unit vector and $\mathbf{R} \in \mathbb{R}^{D \times D}$ be Haar-random orthogonal. Let $\tilde{\mathbf{k}} = \mathbf{R}\hat{\mathbf{k}}$ and partition it into $B$ contiguous subspaces of dimension $m$ ($D = Bm$): $\tilde{\mathbf{k}} = [\tilde{\mathbf{k}}_1; \ldots ; \tilde{\mathbf{k}}_B]$.*

*Define $r_b = \|\tilde{\mathbf{k}}_b\|_2$, $z_b = r_b^2 \in [0, 1]$, and $\mathbf{u}_b = \tilde{\mathbf{k}}_b / r_b \in \mathbb{S}^{m-1}$. Here, $z_b$ is the* subspace energy fraction *(squared radius) after polar decomposition, which we use to design radius quantization centroids (Eq. (13)); $\mathbf{u}_b$ is the corresponding* unit direction *within subspace $b$, whose coordinate-wise distribution in Eq. (12) guides the design of quantization levels for the normalized-and-rotated representation. Then for any fixed $b \in [B]$ and $j \in [m]$:*

$$z_b \sim \mathrm{Beta}\left(\tfrac{m}{2}, \tfrac{D-m}{2}\right), \tag{11}$$

$$(\mathbf{u}_b)_j^2 \sim \mathrm{Beta}\left(\tfrac{1}{2}, \tfrac{m-1}{2}\right). \tag{12}$$

**Implementation.** Both stages are realized with custom CUDA kernels: (i) a collision accumulation kernel and a small-range integer *bucket_topk* kernel for Top-$\beta$ selection, and (ii) a fused reranking kernel for RSQ-IP; details and kernel-level optimizations are deferred to Sec. 4.3 in the Appendix. Additional design details (multi-tier collision weights, offline quantizer derivation, SRHT/Beta-prior justification) are provided in Appendix B.

### 4.2.3. ON-DEMAND UVA FETCH

Finally, the GPU fetches only the selected Top-$k$ KV pairs from CPU memory via UVA, and computes attention (B.3 in Fig. 2). Thus, CPU memory serves as a backing store for capacity, while the retrieval decision path remains GPU-native.

### 4.3. Implementation Optimizations and Complexity

ParisKV relies on a coarse-to-fine retrieval pipeline whose latency is dominated by GPU selection, reranking, and KV

*Table 1.* Hyperparameter configurations across different tasks.

| Dataset | Local | Update | Full-thres. | Max Gen |
|---|---|---|---|---|
| AIME25 | 256 | 512 | 2K | 38.9K |
| MATH500 | 256 | 256 | 1K | 38.9K |
| GPQA-Diamond | 128 | 512 | 2K | 32.8K |
| LongBench_V2 | 256 | 512 | 2K | 1024+512 |
| RULER | 256 | 512 | 2K | 128 |

fetching. We implement four custom CUDA kernels to eliminate sorting overhead, reduce kernel launches, and minimize memory traffic: (i) bucket_topk for integer collision scores, (ii) a parallel collision kernel, (iii) a fused reranking kernel (gather+unpack+score), and (iv) a UVA-based kernel.

**Complexity.** Let $|S|$ be the on-GPU steady/local attention length and $n = |R|$ the retrieval-zone length. Dense scoring over the full cache scales with $\mathcal{O}((|S|+n)D)$ per step. ParisKV keeps dense attention only on $|S|$ and replaces full-cache scoring with a coarse-to-fine pipeline on compact metadata: collision processing scales with $\rho n$, reranking scales with $C = \lceil \beta n \rceil$, and only $k \ll C$ KV vectors are fetched for attention.

## 5. Experimental Evaluation

We evaluate ParisKV on long-generation reasoning (MATH500 (Hendrycks et al., 2021), GPQA-Diamond (Rein et al., 2024), AIME25 (Balunović et al., 2025)) and long-context understanding (LongBench-V2 (Bai et al., 2024)) across three model families (Qwen-3-8B (Qwen Team, 2025), DeepSeek-R1-Llama-8B (Guo et al., 2025), and Qwen3-4B-Thinking-2507 (Qwen Team, 2025)). For fair comparison, we keep the KV-cache configuration in Table 1 consistent across methods. In all experiments, ParisKV uses a *fixed* retrieval budget of top-$K$=100, while PQCache (Zhang et al., 2025b) and MagicPIG (Chen et al., 2024) follow their paper-recommended *budget policies* (PQCache uses 20% compress ratio and MagicPIG uses a dynamic retrieval policy whose effective retrieval size varies with sequence length). Full evaluation protocols, sampling settings, and workload construction details are in Appendix D.

### 5.1. Accuracy Evaluation

We evaluate ParisKV on both long-generation reasoning benchmarks and long-context understanding tasks. In addition to the main comparisons with PQCache and MagicPIG, we evaluate against a broader set of KV-cache retrieval and sparse-attention baselines, including Quest, ShadowKV, FreeKV (Liu et al., 2026), RetroInfer, SOCKET (Joshi et al., 2026), and Twilight (Lin et al., 2025a) where applicable. Across long-generation reasoning, LongBench-V2, and RULER (Hsieh et al., 2024), ParisKV achieves the best overall accuracy among prior KV-cache retrieval/sparse-attention baselines, while often matching or approaching

| Methods | GPQA_dia (pass@1) | MATH500 (pass@1) | AIME_2025 (pass@8) |
|---|---|---|---|
| *Qwen-3-4B* | 64.14 | 88.60 | **86.67** |
| PQCache | 38.38 | 58.80 | 3.33 |
| MagicPIG | 32.32 | 46.40 | 6.67 |
| **ParisKV (Ours)** | **72.22** | **92.80** | **80.00** |
| *DS-R1-Llam-8B* | 49.49 | 80.40 | 50.00 |
| PQCache | 23.81 | 66.75 | 13.30 |
| MagicPIG | 27.78 | 49.00 | 13.30 |
| **ParisKV (Ours)** | **57.07** | **84.40** | **53.30** |
| *Qwen-3-8B* | 54.54 | 87.40 | 83.33 |
| PQCache | 47.85 | 69.21 | 16.67 |
| MagicPIG | 33.84 | 45.80 | 10.00 |
| **ParisKV (Ours)** | **55.05** | **93.00** | **73.33** |

*Table 2.* Accuracy on long-generation tasks: GPQA_diamond (pass@1), MATH500 and AIME_2025 (pass@8).

full attention. The compact main results are shown in Tables 2 and 3; full baseline comparisons are provided in Appendix Tables 5 and 6.

**Long-generation tasks.** Table 2 reports results on GPQA-Diamond (pass@1), MATH500 (pass@1), and AIME25 (pass@8) across three model families. ParisKV consistently outperforms PQCache and MagicPIG, and in 7/9 settings it even **matches or exceeds** full-attention accuracy.

On *Qwen-3-4B*, it achieves 72.22 (GPQA-Diamond) and 92.80 (MATH500), exceeding PQCache by +33.84 and +34.00 points.

**Long-input tasks.** We evaluate long-context understanding on LongBench-V2 (w/ CoT; Appendix D.1). ParisKV consistently improves accuracy over PQCache and MagicPIG, with the strongest gains in the Medium/Long buckets. For example, on DeepSeek-R1-Llama-8B, ParisKV increases the overall score to 28.43 (vs. 19.90 for PQCache and 13.92 for MagicPIG) and achieves 31.11 (Easy) / 30.16 (Hard) on Long inputs (Table 3).

**Summary.** ParisKV improves accuracy in both *long-generation* reasoning and *long-input* understanding settings. Compared to long-generation benchmarks, long-input tasks are typically less prone to retrieval drift because decoding produces far fewer tokens than the prefill length, leaving fewer opportunities for approximation errors to accumulate. Even in this easier regime, ParisKV still consistently outperforms prior KV-cache retrieval baselines on LONGBENCH-V2, including in the Medium/Long buckets. The gains become substantially larger in long-generation reasoning, where drift can compound over tens of thousands of decoding steps: on AIME25, ParisKV improves pass@8 by +40.0 to +76.7 points over PQCache/MagicPIG across model families (Table 2). Overall, these results indicate that ParisKV is robust across long-context regimes, and is especially effective when the decoding horizon is long and retrieval errors would otherwise accumulate.

## 5.2. Efficiency Evaluation

We evaluate ParisKV's decoding efficiency under varying context lengths and batch sizes, compare end-to-end latency with representative KV-cache retrieval systems, and report the overhead of the prefill stage. Since different baselines define and allocate their KV-cache budgets differently, we follow each released implementation's default budget design whenever applicable. For accuracy evaluation, we set the budgets of other baselines to be no smaller than that of ParisKV whenever possible, ensuring that ParisKV is not favored by using a larger retrieval budget. Detailed experimental setup, workloads, metrics, and baseline configurations are provided in Appendix D.1.

**Summary.** ParisKV achieves strong efficiency and scalability for long-context inference. Details of throughput/TPOT/prefill latency, kernel optimizations, and implementation choices are provided in Appendix D.2.

(1) Better batch scaling. Across 64K/128K/256K contexts, ParisKV improves the peak decoding throughput over full attention by $2.1\times \sim 2.8\times$ on both Llama3.1-8B and Qwen3-8B (up to $2.7\times$ on Llama3.1-8B and $2.8\times$ on Qwen3-8B). Moreover, ParisKV sustains significantly larger runnable batch sizes where full attention becomes memory-bounded (e.g., at 128K full attention OOMs at bs$\geq$4 while ParisKV scales to bs=8; at 256K full attention OOMs at bs$\geq$2 while ParisKV scales to bs=5).

(2) Stable decoding latency. ParisKV exhibits smooth TPOT scaling with batch size. For example, on Qwen3-8B at 128K, TPOT increases from 24.32ms/step (bs=1) to 58.92ms/step (bs=8), reducing per-token latency from 24.32ms/token to 7.37 ms/token due to amortization.

(3) Superior performance at extremely long contexts. When full attention becomes infeasible (OOM at 384K even with bs=1), ParisKV remains runnable and scales up to bs=3. Under 128K$\sim$1024K (bs=1), ParisKV substantially outperforms representative KV-retrieval baselines. On Llama3.1-8B, ParisKV achieves up to $44.4\times$ lower decode latency than PQCache (e.g., 49ms/step vs. 2179ms/step at 1024K), and also significantly outperforms MagicPIG with up to $16.9\times$ lower decode latency at 1024K (49ms/step vs. 830ms/step). Similar trends hold for Qwen3-8B, where ParisKV consistently maintains decoding latency in the tens-of-milliseconds range while MagicPIG and PQCache incur substantially higher per-step overheads at long contexts.

## 5.3. Ablation Studies

**Drift mitigation with normalization/rotation + theoretical spherical centroids.** This design substantially improves coarse-stage candidate generation under long decoding and translates into large end-to-end gains: coarse Recall@100 improves from 6% to 16.1%, and final Recall@100 after

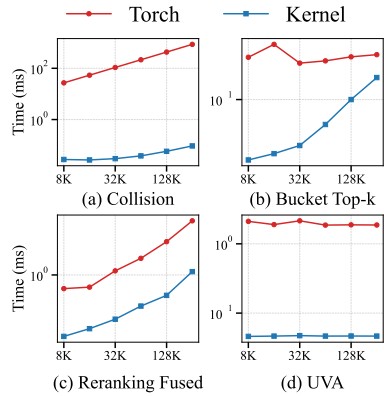

*Figure 6.* Comparison of runtime between Torch and custom kernels.

| Methods | Overall | Short | | Medium | | Long | |
|---|---|---|---|---|---|---|---|
| | | Easy | Hard | Easy | Hard | Easy | Hard |
| *Qwen3-4B* | **25.84** | 27.12 | 16.53 | 36.36 | 25.20 | 26.67 | 28.57 |
| PQCache | 17.91 | 16.95 | 19.00 | 13.60 | 19.00 | 20.00 | 19.05 |
| MagicPIG | 16.70 | 18.64 | 10.74 | 14.77 | 20.47 | 28.89 | 12.70 |
| **ParisKV (Ours)** | 24.60 | 35.59 | 19.49 | 26.14 | 22.05 | 28.89 | 23.81 |
| *Qwen-3-8B* | **33.59** | 50.85 | 34.71 | 32.95 | 25.98 | 37.78 | 28.57 |
| PQCache | 25.50 | 23.70 | 31.60 | 28.20 | 24.00 | 25.00 | 20.00 |
| MagicPIG | 10.34 | 8.47 | 15.70 | 7.95 | 7.87 | 13.33 | 7.94 |
| **ParisKV (Ours)** | 33.07 | 52.54 | 34.71 | 34.09 | 26.77 | 37.21 | 19.67 |
| *DS-R1-Llam-8B* | 13.12 | 18.64 | 15.70 | 12.50 | 8.66 | 11.11 | 14.29 |
| PQCache | 19.90 | 18.60 | 21.50 | 21.60 | 22.20 | 15.60 | 14.30 |
| MagicPIG | 13.92 | 15.25 | 11.57 | 11.36 | 14.17 | 17.78 | 17.46 |
| **ParisKV (Ours)** | **28.43** | 37.29 | 25.62 | 28.41 | 25.20 | 31.11 | 30.16 |

*Table 3.* LongBench_V2 accuracy breakdown by context length (Short/Medium/Long) and difficulty (Easy/Hard).

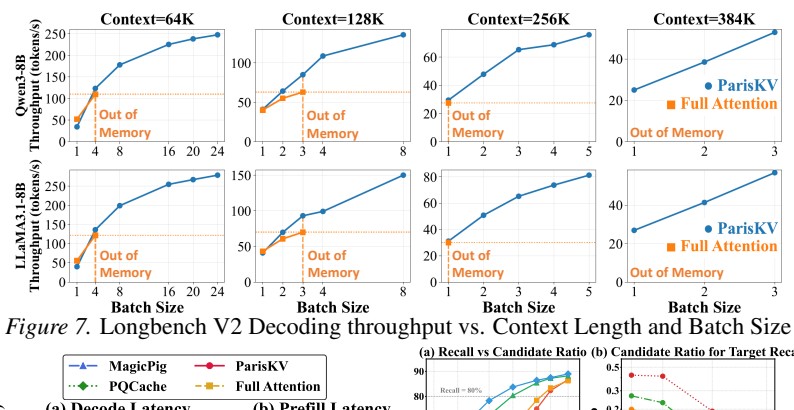

*Figure 7.* Longbench V2 Decoding throughput vs. Context Length and Batch Size

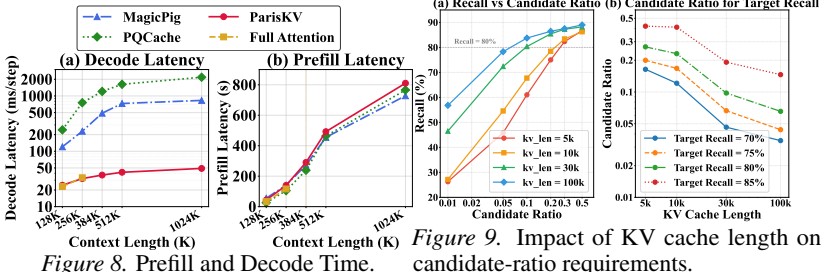

*Figure 8.* Prefill and Decode Time.

*Figure 9.* Impact of KV cache length on candidate-ratio requirements.

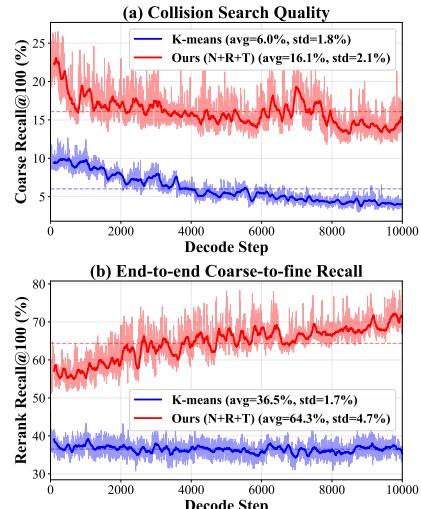

*Figure 10.* Retrieval quality comparisons. (a) Coarse-stage collision quality. (b) End-to-end recall. Ours (N+R+T) denotes normalization + rotation + theoretical centroids.

exact reranking increases from 36.5% to 64.3% (Fig. 10).

# 6. Conclusions

We introduced ParisKV, a drift-robust KV-cache retrieval framework that addresses the fundamental challenges of distribution drift and retrieval latency in long-context LLM inference. Our approach is grounded in a key insight: by transforming keys and queries onto a unit hypersphere via normalization and random orthogonal rotation, we can employ data-independent analytic centroids that remain stable regardless of how the key distribution evolves during decoding. This eliminates the centroid staleness that plagues existing methods and maintains high retrieval recall throughout extended generation.

ParisKV employs a GPU-native coarse-to-fine pipeline combining collision-based pruning with calibrated 4-bit

reranking, and leverages UVA-based offloading for efficient million-token inference with minimal CPU intervention.

Empirically, ParisKV demonstrates substantial improvements over prior KV-cache retrieval methods. On long-generation reasoning tasks, it improves pass@8 accuracy on AIME25 by +40 to +77 points over PQCache and MagicPIG, while matching or exceeding full attention quality in 7 out of 9 settings. The efficiency gains are equally significant: ParisKV achieves up to $2.8\times$ higher decoding throughput than full attention within its runnable range, and reduces decode latency by $17\times$ and $44\times$ compared to MagicPIG and PQCache at million-token scale.

Our results show that carefully designed KV-cache retrieval can simultaneously improve both efficiency and accuracy for long-context inference, even in the presence of drift, opening new possibilities for deploying LLMs in latency-sensitive, long-generation applications.

## Acknowledgements

Supported by EU Horizon project DataGEMS (101188416), and by $\Upsilon\Pi A I \Theta A$ & NextGenerationEU project HARSH ($\Upsilon\Pi 3 T A - 0560901$) that is carried out within the framework of the National Recovery and Resilience Plan "Greece 2.0" with funding from the European Union – NextGenerationEU and National Natural Science Foundation of China (62202450). This work was granted access to the HPC resources of IDRIS under the allocation 2025-A0191012641 made by GENCI.

## Impact Statement

This paper presents work whose goal is to advance efficient long-context inference for large language models. There are many potential societal consequences of our work, none of which we feel must be specifically highlighted here.

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

# A. Additional Related Work

Beyond inference-side KV-cache efficiency, data-centric approaches improve foundation-model training and adaptation efficiency through complementary directions such as synthetic-data pretraining and instruction-data selection (Xie et al., 2026; Lin et al., 2025b; Xie et al., 2025).

## A.1. KV-Cache Quantization

Several recent studies have proposed innovative methods to tackle the challenges of KVCaches quantization while keeping performance degradation minimal.

SmoothQuant(Xiao et al., 2023a) presents an offline approach that migrates the difficulty of quantization to mitigate its associated performance drop, which allows for quantizing the KVCaches to 8-bit precision. This method effectively balances the trade-off between saving memory and preserving model accuracy. Q-Hitter(Zhang et al., 2024) further reduces the KVCaches size by utilizing accumulated attention scores and a "Quantization Friendliness" metric. This method effectively combines sparsity and 8-bit quantization, achieving substantial memory savings while maintaining model performance. KIVI (Liu et al., 2024b) examines the distribution patterns of key and value tokens and introduces an innovative quantization scheme: applying per-token quantization to the key cache and per-channel quantization to the value cache. This method enables an unprecedented 2-bit compression of the KVCaches. QuaRot (Ashkboos et al., 2024) employs a randomized Hadamard transform to induce rotations that suppress outliers, thereby enhancing the model's compatibility with quantization. QuQuant (Lin et al., 2024) detects significant outliers and redistributes them across multiple channels via a series of transformations, which simplifies the quantization process. SpinQuant (Liu et al., 2024c) integrates learnable rotation matrices to optimize the overall network accuracy. KVTuner (Li et al., 2025b) uses offline calibration to determine layer-wise mixed-precision KVCaches configurations, which are directly applied during inference without online adaptation.

## A.2. Vector Search

Vector Approximate Nearest Neighbor search(ANNS) has become a standard building block for KV-Cache Retrieval in large language model inference. By efficiently identifying a small subset of highly relevant tokens, ANNS significantly reduces both memory traffic and computational cost.

Formally, let the KV cache at some layer be a set of key vectors

$$\mathcal{X} = \{x_1, x_2, \ldots, x_N\} \subset \mathbb{R}^d,$$

and let the query vector be

$$q \in \mathbb{R}^d.$$

For a similarity function $s(\cdot, \cdot)$ (e.g., inner product or negative distance), the *exact nearest neighbor* problem is

$$\mathrm{NN}(q) = \arg\max_{x \in \mathcal{X}} s(q, x).$$

The *vector ANN problem* relaxes this to finding $\tilde{x} \in \mathcal{X}$ such that

$$s(q, \tilde{x}) \geq (1 - \varepsilon) \max_{x \in \mathcal{X}} s(q, x),$$

for some approximation factor $\varepsilon > 0$; more practically, one often aims to find a top-$k$ subset whose recall with respect to the exact top-$k$ set is high, while using much less time and memory than exhaustive search.

A rich family of ANNS systems has been developed to support fast, high-accuracy retrieval, including graph-based methods such as HNSW (Malkov & Yashunin, 2018), tree- and partition-based methods such as DB-LSH (Tian et al., 2023), PM-LSH (Zheng et al., 2020), DET-LSH (Wei et al., 2024; 2026), MESSI (Peng et al., 2020; 2021), and SOFA (Schäfer et al., 2025), quantization-based methods such as LVQ (Aguerrebere et al., 2023), SAQ (Li et al., 2025a), RabitQ (Gao & Long, 2024) and SuCo (Wei et al., 2025), and hierarchical methods such as LSH-APG (Zhao et al., 2023), SymphonyQG (Gou et al., 2025) and ELPIS (Azizi et al., 2023). These techniques form the algorithmic foundation of modern KV-cache retrieval systems operating under strict latency and memory constraints while preserving attention accuracy.

# B. Additional Details of On-GPU Retrieval (Coarse-to-Fine)

**Setup.** Let $\hat{\mathbf{k}} = \mathbf{k}/\|\mathbf{k}\|_2$ and $\hat{\mathbf{q}} = \mathbf{q}/\|\mathbf{q}\|_2$. We apply an orthogonal rotation $\mathbf{R}$ (SRHT in implementation) and partition into $B$ contiguous subspaces: $\tilde{\mathbf{k}} = \mathbf{R}\hat{\mathbf{k}} = [\tilde{\mathbf{k}}_1; \ldots; \tilde{\mathbf{k}}_B]$, $\tilde{\mathbf{q}} = \mathbf{R}\hat{\mathbf{q}} = [\tilde{\mathbf{q}}_1; \ldots; \tilde{\mathbf{q}}_B]$, where $\tilde{\mathbf{k}}_b, \tilde{\mathbf{q}}_b \in \mathbb{R}^m$ and $D = Bm$. In each subspace $b$, each key is assigned to a centroid $\mathbf{c}_{i,b}$ (learned on prefill stage). At decoding time, we compute a cheap proxy score per centroid (e.g., $\tilde{\mathbf{q}}_b^\top \mathbf{c}$) and rank keys by their centroid score.

## B.1. Offline Quantizer Derivation from Rotation-Induced Priors

### B.1.1. ROTATION-INDUCED BETA PRIORS

**Proposition B.1** (Rotation-induced Beta priors for subspaces). *Let $\hat{\mathbf{k}} \in \mathbb{S}^{D-1}$ be any unit vector and $\mathbf{R} \in \mathbb{R}^{D \times D}$ be Haar-random orthogonal. Let $\tilde{\mathbf{k}} = \mathbf{R}\hat{\mathbf{k}}$ and partition it into $B$ contiguous subspaces of dimension $m$ ($D = Bm$): $\tilde{\mathbf{k}} = [\tilde{\mathbf{k}}_1; \ldots; \tilde{\mathbf{k}}_B]$. Define $r_b = \|\tilde{\mathbf{k}}_b\|_2$, $z_b = r_b^2 \in [0, 1]$, and $\mathbf{u}_b = \tilde{\mathbf{k}}_b/r_b \in \mathbb{S}^{m-1}$. Then for any fixed $b \in [B]$ and $j \in [m]$:*

$$z_b \sim \text{Beta}\left(\tfrac{m}{2}, \tfrac{D-m}{2}\right), \tag{13}$$

$$(\mathbf{u}_b)_j^2 \sim \text{Beta}\left(\tfrac{1}{2}, \tfrac{m-1}{2}\right). \tag{14}$$

*Proof sketch.* Since $\mathbf{R}$ is Haar-random, $\tilde{\mathbf{k}}$ is uniform on $\mathbb{S}^{D-1}$. Using $\tilde{\mathbf{k}} \overset{d}{=} \mathbf{g}/\|\mathbf{g}\|_2$ with $\mathbf{g} \sim \mathcal{N}(0, \mathbf{I}_D)$, each block energy ratio $z_b = \|\mathbf{g}_b\|_2^2/\|\mathbf{g}\|_2^2$ becomes a ratio of independent chi-square variables, yielding (13). Moreover, $\mathbf{u}_b = \mathbf{g}_b/\|\mathbf{g}_b\|_2$ is uniform on $\mathbb{S}^{m-1}$ and $(\mathbf{u}_b)_j^2$ is again a chi-square ratio, yielding (14).

**Remark (SRHT vs. Haar).** Our analysis assumes Haar-random $\mathbf{R}$. In practice, we implement $\mathbf{R}$ using SRHT, a fast randomized orthogonal transform that empirically induces near-isotropic coordinate statistics. We therefore reuse the same analytic priors as a stable approximation under this near-isotropy assumption.

### B.1.2. COORDINATE-MAGNITUDE QUANTIZER FOR 4-BIT DIRECTION CODES

RSQ-IP quantizes $X = |(\mathbf{u}_b)_j|$ (with an additional sign bit). From (12), $Y = (\mathbf{u}_b)_j^2 \sim \text{Beta}\left(\tfrac{1}{2}, \tfrac{m-1}{2}\right)$, hence $X = \sqrt{Y} \in [0, 1]$ has an analytic, data-independent target distribution. We compute shared 3-bit thresholds $\boldsymbol{\tau} = (\tau_1, \ldots, \tau_7)$ and reconstruction levels $a[0], \ldots, a[7]$ by running Lloyd–Max scalar quantization on the density of $X$ (offline, once), and share them across all layers/subspaces.

**Lloyd–Max updates.** Given thresholds, reconstruction values are conditional means $a[t] \leftarrow \mathbb{E}[X \mid \tau_t \leq X < \tau_{t+1}]$, and given reconstruction values, thresholds are midpoints $\tau_t \leftarrow (a[t-1] + a[t])/2$ for $t = 1, \ldots, 7$. We iterate until convergence (offline). Because the target distribution depends only on $m$, this quantizer is data-independent and stable under decoding drift.

### B.1.3. (OPTIONAL) RADIUS/ENERGY QUANTIZATION

Rotation and partitioning also yield a stable prior for subspace energy. Define $z_{i,b} \triangleq r_{i,b}^2 = \|\tilde{\mathbf{k}}_{i,b}\|_2^2 \in [0, 1]$. From (13), $z_{i,b} \sim \text{Beta}\left(\tfrac{m}{2}, \tfrac{D-m}{2}\right)$, which enables data-independent radius centers via Lloyd–Max quantization of $z$ (followed by square-roots for $r$).

In our final system, we do *not* quantize radii for reranking: we retain exact $r_{i,b}$ and absorb it into the precomputed weight $w_{i,b} = \|\mathbf{k}_i\|_2 \, r_{i,b}/\alpha_{i,b}$. We also set the number of radius levels to $K_r = 1$ in coarse retrieval because finer radius binning provides marginal recall gains but increases memory and update overhead.

## B.2. On-GPU Retrieval (Coarse-to-Fine).

At each decoding step, ParisKV performs a two-stage GPU pipeline: *Stage I* uses inexpensive *direction-only* collisions across subspaces to prune the search space to a small candidate set (typically top-$\beta$ fraction), and *Stage II* reranks candidates with RSQ-IP by estimating the raw attention score $\langle \mathbf{k}, \mathbf{q} \rangle$ from the compact 4-bit summaries (no full-precision keys needed during reranking). Both stages are implemented with custom CUDA kernels (bucket top-k kernel, collision kernel, reranking fused kernel, UVA kernel), which we detail in Section 4.3.

### B.2.1. COARSE CANDIDATE GENERATION (COLLISION RATIO AND MULTI-TIER WEIGHTS).

We aim to quickly prune the full key set to a small candidate set while remaining robust to decoding drift. ParisKV forms a coarse score by accumulating subspace-wise *collisions*. In each subspace, we rank keys by a cheap similarity between the query and the key's assigned centroid (e.g., $q^\top c$), and only the top-$\rho$ fraction participates in scoring; keys outside this top-$\rho$ set receive zero bonus in that subspace. Within the collided region, we apply *multi-tier* bonuses rather than a uniform $+1$.

A binary (0/1) collision yields a low-resolution total score (e.g., $[0, B]$ for $B=16$), so many keys end up with identical accumulated counts. This coarseness is problematic for two reasons: (i) it cannot distinguish keys that are highly aligned with the query in some subspaces but weakly aligned in others, weakening the ranking signal from multi-subspace accumulation; and (ii) it makes Top-$\beta$ truncation unstable, since the cutoff often contains a large group of keys with the same score, forcing the candidate set to exceed the intended budget and causing candidate sizes to vary across heads/steps.

**Tier definition.** Within the top-$\rho$ fraction in each subspace, we further split keys into $L$ percentile tiers and assign tier weights. In our implementation, we use $L = 6$ tiers with weights and cutoffs:

$$\text{weights } \{6, 5, 4, 3, 2, 1\}, \quad \text{percentiles } \{5\%, 15\%, 30\%, 50\%, 75\%, 100\%\} \text{ (within top-}\rho\text{)}.$$

Thus each subspace contributes an integer bonus in $\{0, 1, \ldots, 6\}$ and the summed score range becomes $[0, 6B]$ (e.g., $[0, 96]$ when $B=16$), yielding a sharper and more stable Top-$\beta$ cutoff.

**Coarse score.** Let $T_b(\rho)$ be the set of keys in the top-$\rho$ fraction under centroid proxy in subspace $b$. Let $\ell(i, b) \in \{1, \ldots, L\}$ denote the tier index for key $i$ in subspace $b$ (defined only if $i \in T_b(\rho)$). The collision score is

$$S_i \triangleq \sum_{b=1}^{B} \mathbb{I}[i \in T_b(\rho)] \cdot w_{\ell(i,b)}, \tag{15}$$

where $w_\ell$ is the tier weight.

After summing across $B$ subspaces, we select the top-$\beta$ fraction of keys as candidates (typically $\beta=5\%$–$10\%$). In practice, we choose $\rho \geq \beta$ so that each subspace scores at least as many keys as the final candidate budget. Otherwise, too few keys receive non-zero bonuses in that subspace, and the accumulated collision score becomes overly sparse, weakening the ability to reliably identify the Top-$\beta$ candidates. and ensure the candidate set is comfortably larger than the final Top-$k$ (e.g., $k=100$) used in Stage II. Since $k$ is fixed, the optimal $\beta$ depends strongly on the KV length: shorter KV often requires a larger $\beta$ to ensure enough candidates can cover the true Top-$k$, whereas longer KV allows a smaller $\beta$ to reduce reranking cost.

**Top-$\beta$ selection with small-range integer scores** We implement collision accumulation with a dedicated *Collision Kernel*, and use a specialized *bucket_topk* kernel to select Top-$\beta$ candidates from small-range integer scores without sorting. We avoid sorting all $N$ keys by exploiting that $S_i$ is small-range integer. We use a histogram-and-prefix-scan `bucket_topk` kernel: (i) build a histogram of scores, (ii) find the threshold score yielding $\lceil \beta N \rceil$ items, (iii) compact indices whose scores exceed the threshold; ties at the threshold are handled by deterministic truncation. This yields predictable candidate sizes and avoids expensive global sort.

### B.2.2. RSQ-IP RERANKING FROM 4-BIT KEY SUMMARIES

**Goal.** Given Stage I candidates, RSQ-IP reranks them by estimating the raw attention score $\langle \mathbf{k}_i, \mathbf{q} \rangle$ directly from compact 4-bit key summaries, since full-precision keys are offloaded and unavailable during reranking. It combines (i) data-independent low-bit quantization with theoretically derived levels and (ii) an alignment correction to reduce quantization bias.

**Notation (radius–direction decomposition).** After normalization and rotation, we partition each key into $B$ subspaces. For key $i$ and subspace $b$, define the subspace radius and unit direction $r_{i,b} \triangleq \|\tilde{\mathbf{k}}_{i,b}\|_2$ and $\mathbf{u}_{i,b} \triangleq \tilde{\mathbf{k}}_{i,b}/r_{i,b} \in \mathbb{S}^{m-1}$. Then the cosine between unit vectors decomposes as

$$\langle \hat{\mathbf{k}}_i, \hat{\mathbf{q}} \rangle = \sum_{b=1}^{B} r_{i,b} \langle \mathbf{u}_{i,b}, \tilde{\mathbf{q}}_b \rangle. \tag{16}$$

**Step 1: 4-bit direction quantization (1-bit sign + 3-bit magnitude).** We encode each coordinate of $\mathbf{u}_{i,b} \in \mathbb{R}^m$ using a sign bit and a 3-bit magnitude bin:

$$s_{i,b,j} \triangleq \text{sign}((\mathbf{u}_{i,b})_j) \in \{+1, -1\}, \qquad t_{i,b,j} \triangleq \text{bucketize}_\tau(|(\mathbf{u}_{i,b})_j|) \in \{0, \ldots, 7\}. \tag{17}$$

Given reconstruction levels $a[0], \ldots, a[7]$, we reconstruct a quantized direction $\mathbf{v}_{i,b}$ by $(\mathbf{v}_{i,b})_j \triangleq s_{i,b,j} \, a[t_{i,b,j}]$. We derive a *data-independent* shared $(\boldsymbol{\tau}, a[\cdot])$ offline from rotation-induced priors (Appendix B.1.1).

**Step 2: Alignment-corrected subspace estimator.** For each subspace, we approximate the inner product $\langle \mathbf{u}_{k,b}, \tilde{\mathbf{q}}_b \rangle$ using a compact quantized direction code $\mathbf{v}_{k,b}$ together with a *correction factor* (a.k.a. alignment factor)

$$\alpha_{k,b} \triangleq \langle \mathbf{v}_{k,b}, \mathbf{u}_{k,b} \rangle. \tag{18}$$

Specifically, RSQ-IP employs the approximation

$$\langle \mathbf{u}_{k,b}, \tilde{\mathbf{q}}_b \rangle \approx \frac{\langle \mathbf{v}_{k,b}, \tilde{\mathbf{q}}_b \rangle}{\alpha_{k,b}}. \tag{19}$$

This form is inspired by RaBitQ's (Gao & Long, 2024) alignment-corrected estimator for *unit-vector* inner products, but here it is instantiated *subspace-wise* and embedded into a rotation–decomposition pipeline tailored to KV cache.

**Step 3: Cosine estimator** Combining (16) and (19), we obtain a cosine-similarity estimator:

$$\widehat{\langle \hat{\mathbf{k}}, \hat{\mathbf{q}} \rangle} \triangleq \sum_{b=1}^{B} r_{k,b} \frac{\langle \mathbf{v}_{k,b}, \tilde{\mathbf{q}}_b \rangle}{\alpha_{k,b}}. \tag{20}$$

**Step 4: Recovering the *raw* inner product for KV-cache reranking.** The pre-softmax attention score uses the unnormalized inner product:

$$\langle \mathbf{k}, \mathbf{q} \rangle = \|\mathbf{k}\|_2 \, \|\mathbf{q}\|_2 \, \langle \hat{\mathbf{k}}, \hat{\mathbf{q}} \rangle. \tag{21}$$

Plugging (20) into (21) yields the RSQ-IP estimator of the raw score:

$$\widehat{\langle \mathbf{k}, \mathbf{q} \rangle} \triangleq \|\mathbf{q}\|_2 \sum_{b=1}^{B} \left( \|\mathbf{k}\|_2 \, r_{k,b} \right) \frac{\langle \mathbf{v}_{k,b}, \tilde{\mathbf{q}}_b \rangle}{\alpha_{k,b}}. \tag{22}$$

For efficiency, we precompute and store a per-subspace *scaling factor*

$$w_{i,b} \triangleq \|\mathbf{k_i}\|_2 \frac{r_{i,b}}{\alpha_{i,b}}, \tag{23}$$

which absorbs all key-dependent terms in the RSQ-IP estimator. As a result, online reranking only requires computing the lightweight dot product $\langle \mathbf{v}_{i,b}, \tilde{\mathbf{q}}_b \rangle$ and accumulating

$$\widehat{\langle \mathbf{k}, \mathbf{q} \rangle} = \|\mathbf{q}\|_2 \sum_{b=1}^{B} w_{i,b} \langle \mathbf{v}_{i,b}, \tilde{\mathbf{q}}_b \rangle. \tag{24}$$

# C. Observations

# D. Experiment

### D.1. Experimental Setup and Baseline Configurations

We evaluate ParisKV against representative KV-cache retrieval and sparse-attention baselines, including Quest, MagicPIG, ShadowKV, FreeKV (Liu et al., 2026), RetroInfer, PQCache, SOCKET (Joshi et al., 2026), and Twilight (Lin et al., 2025a). Different baselines define their effective KV-cache budgets differently. Some methods allocate tokens across local windows, sink tokens, outlier chunks, or sparse retrieval budgets, while others expose a single top-$k$ or page-level budget. Therefore, we follow the released implementation of each baseline and choose configurations that are at least as favorable as ParisKV whenever possible. In particular, for accuracy evaluation, we ensure that the effective budget of each baseline is no smaller than that of ParisKV, so that ParisKV is not advantaged by a larger retrieval budget.

**MagicPIG.** The original MagicPIG constructs LSH indices only over keys generated during the prefill phase and uses them to select top-$k$ candidates. Keys generated during the decode phase are still fully included in the attention computation. On reasoning tasks such as AIME2025, which typically have short inputs but long outputs, this design leads to little practical reduction in computation compared with full attention. To ensure a fair comparison in long-generation settings, we extend MagicPIG by treating keys from both the prefill and decode phases as the candidate set for top-$k$ selection.

| Symbol | Meaning |
|---|---|
| $D$ | KV dimension (256) |
| $B$ | #subspaces (16) |
| $m$ | subspace dim ($m=D/B=8$) |
| $\mathbf{k}, \mathbf{q} \in \mathbb{R}^D$ | key / query |
| $\|\mathbf{k}\|_2, \|\mathbf{q}\|_2$ | $\ell_2$ norms |
| $\hat{\mathbf{k}}, \hat{\mathbf{q}}$ | normalized vectors |
| $\mathbf{R} \in \mathbb{R}^{D \times D}$ | SRHT rotation |
| $\tilde{\mathbf{k}}, \tilde{\mathbf{q}}$ | rotated vectors $\mathbf{R}\hat{\mathbf{k}}, \mathbf{R}\hat{\mathbf{q}}$ |
| $\tilde{\mathbf{k}}_b, \tilde{\mathbf{q}}_b \in \mathbb{R}^m$ | $b$-th subspace subvectors |
| $r_{k,b}, r_{q,b}$ | subspace radii $\|\tilde{\mathbf{k}}_b\|_2, \|\tilde{\mathbf{q}}_b\|_2$ |
| $\mathbf{u}_{k,b}, \mathbf{u}_{q,b}$ | subspace directions $\tilde{\mathbf{k}}_b/r_{k,b}, \tilde{\mathbf{q}}_b/r_{q,b}$ |
| $s_{k,b,j}$ | sign code of $(\mathbf{u}_{k,b})_j$ |
| $t_{k,b,j}$ | 3-bit magnitude bin of $|(\mathbf{u}_{k,b})_j|$ |
| $\boldsymbol{\tau}, a[\cdot]$ | magnitude thresholds / centers |
| $\mathbf{v}_{k,b} \in \mathbb{R}^m$ | reconstructed 4-bit direction |
| $\alpha_{k,b}$ | correction (alignment) factor |
| $w_{k,b}$ | stored weight $\|\mathbf{k}\|_2\, r_{k,b}/\alpha_{k,b}$ |
| $\langle \mathbf{k}, \mathbf{q} \rangle$ | pre-softmax attention score |
| $\widehat{\langle \mathbf{k}, \mathbf{q} \rangle}$ | RSQ-IP estimate |
| $\rho, \beta$ | collision ratio / candidate ratio |
| $k$ | final top-$k$ |

*Table 4.* Notation for RSQ-IP and subspace-collision.

**ShadowKV.** We evaluate ShadowKV under two configurations. The first setting uses $local\_chunk = 32$ with $chunk\_size = 8$, corresponding to 256 local tokens, $outlier\_chunk = 48$ with $chunk\_size = 8$, corresponding to 384 outlier tokens, and a $sparse\_budget$ of 512 tokens for top-$k$ retrieval. This gives a total per-step budget of 1152 tokens.

The second setting uses $local\_chunk = 4$ with $chunk\_size = 8$, corresponding to 32 local tokens, $outlier\_chunk = 48$ with $chunk\_size = 8$, corresponding to 384 outlier tokens, and a $sparse\_budget$ of 104 tokens for top-$k$ retrieval. This gives a total per-step budget of 520 tokens.

**FreeKV.** For FreeKV, we use $sink = 64$, $local = 256$, and a page budget of $4 \times 32 = 128$ tokens, resulting in a total effective budget of 448 tokens.

**Notation.** NA denotes that a setting is unsupported by the released implementation. OOM denotes out-of-memory.

### D.1.1. ACCURACY EVALUATION

Table 5 reports the full LongBench-V2 comparison with additional baselines that are omitted from the compact main-text table. Table 6 reports the RULER breakdown at 128K context. On LongBench-V2, the additional baselines exhibit uneven behavior across models and context buckets: RetroInfer is competitive on Qwen3-4B, Quest is strong on several Qwen-3-8B splits, and SOCKET/Twilight improve some DS-R1-Llama-8B cases. Nevertheless, ParisKV provides the best overall accuracy on the DS-R1-Llama-8B setting and remains among the top methods on Qwen models while using the same fixed Top-100 retrieval budget as in the main experiments. This indicates that the gains are not limited to comparisons with PQCache and MagicPIG, but persist against recent sparse-attention and KV-retrieval baselines.

RULER (Hsieh et al., 2024) further stresses retrieval with synthetic long-context tasks at 128K tokens. ParisKV attains the highest average score among sparse/retrieval methods (83.49), outperforming ShadowKV (80.29), Quest (79.34), SOCKET (78.92), Twilight (77.60), RetroInfer (77.45), MagicPIG (70.49), and PQCache (33.76). The improvement is especially visible on multi-key retrieval tasks such as `mk2_niah`, where ParisKV closes much of the gap to full attention while preserving a sparse retrieval budget.

### D.1.2. EFFICIENCY SETUP AND LATENCY MEASUREMENTS

Table 7 reports the new long-context speed measurements.

---

**Algorithm 1** ParisKV Two-Stage KV-Cache Retrieval (Per Head)

---

1: **Input:** KV keys $\{\mathbf{k}_i\}_{i=1}^n$, query $\mathbf{q}$, params $(B, m)$, collision ratio $\rho$, candidate ratio $\beta$, final top-$k$
2: **Output:** Indices $\mathcal{I}_{\text{top-}k}$
3: **Encode (prefill / streaming / offload).**
4: **for** $i = 1$ **to** $n$ **do**
5: $\quad \hat{\mathbf{k}}_i \leftarrow \mathbf{k}_i / \|\mathbf{k}_i\|_2$; $\tilde{\mathbf{k}}_i \leftarrow \mathbf{R}\hat{\mathbf{k}}_i$
6: $\quad$ split $\tilde{\mathbf{k}}_i \rightarrow \{\tilde{\mathbf{k}}_{i,b}\}_{b=1}^B$
7: $\quad$ **for** $b = 1$ **to** $B$ **do**
8: $\quad\quad r_{i,b} \leftarrow \|\tilde{\mathbf{k}}_{i,b}\|_2$; $\mathbf{u}_{i,b} \leftarrow \tilde{\mathbf{k}}_{i,b}/r_{i,b}$
9: $\quad\quad$ store coarse direction code $\omega_{i,b}$; store 4-bit code $\mathbf{v}_{i,b}$, $\alpha_{i,b}$, weight $w_{i,b}$
10: $\quad$ **end for**
11: **end for**
12: *All selection runs on GPU; CPU is a backing store for full KV tensors.*
13: **Decode (online query).**
14: $\hat{\mathbf{q}} \leftarrow \mathbf{q}/\|\mathbf{q}\|_2$; $\tilde{\mathbf{q}} \leftarrow \mathbf{R}\hat{\mathbf{q}}$
15: split $\tilde{\mathbf{q}} \rightarrow \{\tilde{\mathbf{q}}_b\}_{b=1}^B$
16: **Stage I: coarse candidate generation (direction-only).**
17: initialize collision counter $S[i] \leftarrow 0$ for all $i \in [n]$
18: **for** $b = 1$ **to** $B$ **do**
19: $\quad$ select top clusters covering ratio $\rho$; update $S[i]$ by tier-weighted collisions
20: **end for**
21: $\mathcal{C} \leftarrow$ top-$\lceil \beta n \rceil$ indices by $S[i]$
22: **Stage II: RSQ-IP reranking on $\mathcal{C}$.**
23: **for** $i \in \mathcal{C}$ **do**
24: $\quad \widehat{s}_i \leftarrow \|\mathbf{q}\|_2 \sum_{b=1}^B w_{i,b} \langle \mathbf{v}_{i,b}, \tilde{\mathbf{q}}_b \rangle$
25: **end for**
26: **Return** top-$k$ indices in $\mathcal{C}$ by $\widehat{s}_i$

---

**Setup.** We conduct experiments on Llama3.1-8B and Qwen3-8B using workloads derived from LongBench v2. Specifically, we select the first sample from the easy + long subset. Since samples in the long subset typically exceed 128K tokens, we truncate each input to 128K tokens to ensure consistent and reproducible comparisons. For a batch size of bs, we construct each batch using the first bs samples from the same subset. Unless otherwise specified, we set sink (steady tokens) to 128 and the local window to 512.

**Workloads**. We evaluate batch scaling across four context lengths: 64K / 128K / 256K / 384K. In addition, to examine scalability under extremely long contexts, we report results at 512K / 1024K with bs = 1.

**Metrics**. We report three primary efficiency metrics: (1) Prefill latency (s); (2) Decode latency / TPOT (ms/step); and (3) Decoding throughput (tokens/s) (measured during the decoding stage). We also report normalized per-token latency, computed as (decode ms/step) ÷ batch size, to characterize the amortization effect from larger batches.

**Baselines**. We use FlashAttention-2 as the implementation of the full-attention baseline. Both MagicPIG and PQCache follow their original algorithms and recommended hyperparameters from the respective papers. For MagicPIG, we keep L, K, and the retrieval strategy unchanged, and only align sink/local with ParisKV for fair comparison. For PQCache, we adopt the recommended top-k budget of 20% to enable an efficiency comparison under a reasonably strong (and typically more accurate) configuration. This choice is motivated by the goal of comparing speed under comparable accuracy targets; however, we find that PQCache still falls short of ParisKV's accuracy even at this budget.

Note that in the original design of MagicPIG, the retrieval region mainly applies during prefill, while newly generated tokens during decoding still participate in full attention. For our evaluated workloads with long inputs and short generation, this design choice does not alter the fundamental cost structure of prefill/decoding.

Since PQCache does not support throughput evaluation under multi-batch settings, for batch scaling experiments we compare ParisKV only with full attention, while for single-batch long-context comparisons we evaluate ParisKV, MagicPIG, and PQCache together.

| Methods | Overall | Short | | Medium | | Long | |
|---|---|---|---|---|---|---|---|
| | | Easy | Hard | Easy | Hard | Easy | Hard |
| *Qwen3-4B* | **25.84** | 27.12 | 16.53 | 36.36 | 25.20 | 26.67 | 28.57 |
| PQCache | 17.91 | 16.95 | 19.00 | 13.60 | 19.00 | 20.00 | 19.05 |
| MagicPIG | 16.70 | 18.64 | 10.74 | 14.77 | 20.47 | 28.89 | 12.70 |
| ShadowKV | 16.30 | 13.60 | 14.80 | 22.20 | 9.90 | 21.30 | 19.00 |
| FreeKV | 19.68 | 20.34 | 13.22 | 21.59 | 24.41 | 22.22 | 17.46 |
| Quest | 19.12 | 32.00 | 16.18 | 21.43 | 12.50 | 21.05 | 24.24 |
| RetroInfer | 23.69 | 34.50 | 31.70 | 22.70 | 22.20 | 13.60 | 9.70 |
| SOCKET | 17.93 | 20.00 | 22.06 | 16.67 | 12.50 | 15.79 | 21.21 |
| Twilight | 19.12 | 28.00 | 14.71 | 21.43 | 15.62 | 21.05 | 24.24 |
| **ParisKV (Ours)** | 24.60 | 35.59 | 19.49 | 26.14 | 22.05 | 28.89 | 23.81 |
| *Qwen-3-8B* | **33.59** | 50.85 | 34.71 | 32.95 | 25.98 | 37.78 | 28.57 |
| PQCache | 25.50 | 23.70 | 31.60 | 28.20 | 24.00 | 25.00 | 20.00 |
| MagicPIG | 10.34 | 8.47 | 15.70 | 7.95 | 7.87 | 13.33 | 7.94 |
| ShadowKV | 15.90 | 40.70 | 23.10 | 6.80 | 13.40 | 6.70 | 3.20 |
| FreeKV | 15.31 | 28.81 | 24.79 | 15.91 | 11.81 | 6.70 | 3.20 |
| Quest | 23.90 | 36.00 | 25.00 | 28.57 | 15.62 | 31.58 | 18.18 |
| RetroInfer | 20.48 | 44.80 | 36.70 | 20.50 | 7.90 | 4.50 | 3.20 |
| SOCKET | 17.93 | 24.00 | 22.06 | 14.29 | 20.31 | 10.53 | 9.09 |
| Twilight | 20.32 | 36.00 | 26.47 | 19.05 | 12.50 | 15.79 | 15.15 |
| **ParisKV (Ours)** | 33.07 | 52.54 | 34.71 | 34.09 | 26.77 | 37.21 | 19.67 |
| *DS-R1-Llam-8B* | 13.12 | 18.64 | 15.70 | 12.50 | 8.66 | 11.11 | 14.29 |
| PQCache | 19.90 | 18.60 | 21.50 | 21.60 | 22.20 | 15.60 | 14.30 |
| MagicPIG | 13.92 | 15.25 | 11.57 | 11.36 | 14.17 | 17.78 | 17.46 |
| ShadowKV | 14.51 | 18.60 | 23.10 | 13.60 | 14.20 | 2.20 | 4.80 |
| FreeKV | 17.50 | 44.07 | 30.58 | 9.09 | 8.66 | 6.67 | 4.76 |
| Quest | 23.51 | 24.00 | 27.94 | 28.57 | 14.06 | 26.32 | 24.24 |
| RetroInfer | 15.66 | 27.60 | 20.00 | 20.50 | 9.50 | 4.50 | 9.70 |
| SOCKET | 21.51 | 20.00 | 14.71 | 30.95 | 21.88 | 15.79 | 27.27 |
| Twilight | 20.72 | 20.00 | 14.71 | 26.19 | 23.44 | 15.79 | 24.24 |
| **ParisKV (Ours)** | **28.43** | 37.29 | 25.62 | 28.41 | 25.20 | 31.11 | 30.16 |

*Table 5.* Full LongBench-V2 accuracy breakdown by context length and difficulty, including additional baselines.

| Methods | s1_niah | s2_niah | mk1_niah | mk2_niah | mv_niah | mq_niah | fwe | qa_1 | qa_2 | vt | Avg. |
|---|---|---|---|---|---|---|---|---|---|---|---|
| *Llama-3.1-8B* | 100.00 | 100.00 | 96.88 | 89.58 | 98.44 | 99.74 | 71.18 | 86.46 | 51.04 | 47.92 | 84.12 |
| PQCache | 5.21 | 50.00 | 45.83 | 30.21 | 16.67 | 20.31 | 55.21 | 68.75 | 40.62 | 4.79 | 33.76 |
| MagicPIG | 100.00 | 94.79 | 83.33 | 34.38 | 73.44 | 78.12 | 70.08 | 73.96 | 42.71 | 53.33 | 70.49 |
| ShadowKV | 100.00 | 98.96 | 96.88 | 73.96 | 89.06 | 96.35 | 61.11 | 81.25 | 50.00 | 55.42 | 80.29 |
| SOCKET | 100.00 | 100.00 | 96.88 | 72.92 | 92.71 | 98.18 | 33.33 | 80.21 | 46.88 | 68.13 | 78.92 |
| Twilight | 98.96 | 100.00 | 95.83 | 76.04 | 90.36 | 97.92 | 57.29 | 80.21 | 48.96 | 74.17 | 77.60 |
| RetroInfer | 95.83 | 98.96 | 95.83 | 55.21 | 94.53 | 97.40 | 61.11 | 78.12 | 42.71 | 54.79 | 77.45 |
| Quest | 99.00 | 98.96 | 94.79 | 64.58 | 83.59 | 94.79 | 62.85 | 79.17 | 44.79 | 70.83 | 79.34 |
| **ParisKV** | 100.00 | 100.00 | 96.88 | 82.01 | 95.83 | 98.70 | 56.25 | 84.38 | 48.96 | 71.87 | 83.49 |

*Table 6.* **Model Accuracy on RULER with 128K Context.** Accuracy comparison across different methods on the full RULER breakdown.

### D.2. Efficiency Evaluation

**Throughput**. Fig. 7 reports the decoding throughput of ParisKV and full attention under varying context lengths and batch sizes. Overall, ParisKV exhibits substantially better batch scalability across all comparable settings. As the batch size increases, ParisKV continues to improve throughput, whereas full attention quickly hits out-of-memory (OOM), due to GPU memory constraints, failing to scale further.

For context lengths of 64K / 128K / 256K, ParisKV achieves clear throughput advantages in the scalable batch region. Specifically, on Llama3.1-8B, ParisKV improves the peak decoding throughput over full attention by 2.3× / 2.1× / 2.7×, respectively. On Qwen3-8B, the corresponding improvements are 2.3× / 2.2× / 2.8×. When the context length further increases to 384K, full attention runs into OOM even at bs=1, while ParisKV remains stable and scales up to bs=3, demonstrating its capability for longer-context inference.

**Kernel Optimization.** Our throughput gains are further supported by GPU kernel optimizations (Fig. 6). Compared with Torch implementations, our collision operator achieves a $9.2\times$ speedup at KV Len $= 256$K, UVA-based KV fetching is about $40\times$ faster, and the fused reranking kernel provides a consistent 3–4× speedup. The bucket-based Top-$k$ kernel reaches up to $9.4\times$ speedup on short contexts (e.g., 16K) and remains competitive as context grows.

Finally, ParisKV requires periodically updating newly generated keys into the codebook, which introduces non-trivial overhead if performed every decoding step. We therefore amortize this cost by performing codebook updates only once every fixed interval (e.g., every 256/512 decoding steps). Note that at 384K, ParisKV could potentially scale to even larger batch sizes; in our current setup, scalability is primarily bounded by the available CPU memory (167GB free), which limits further offloading of KV cache.

| Input Len. | Quest Pre. | Quest Dec. | Twilight Pre. | Twilight Dec. | RetroInfer Pre. | RetroInfer Dec. | Full Pre. | Full Dec. | SOCKET Pre. | SOCKET Dec. | MagicPIG Pre. | MagicPIG Dec. | PQCache Pre. | PQCache Dec. | ShadowKV Pre. | ShadowKV Dec. | ParisKV Pre. | ParisKV Dec. |
|---|---|---|---|---|---|---|---|---|---|---|---|---|---|---|---|---|---|---|
| 128K | 37.28 | 20.78 | 34.56 | 1106.14 | 35.74 | 29.14 | 33.30 | 23.13 | 38.29 | 225.00 | 55.10 | 120.57 | 25.50 | 243.91 | 51.77 | 18.76 | **43.00** | **24.42** |
| 256K | 117.71 | 46.30 | OOM | OOM | 126.71 | 37.53 | 116.20 | 33.29 | OOM | OOM | 141.10 | 230.24 | 104.70 | 754.76 | 152.12 | 19.11 | **139.50** | **32.16** |
| 384K | OOM | OOM | OOM | OOM | 274.85 | 33.31 | OOM | OOM | OOM | OOM | 272.80 | 487.77 | 238.80 | 1207.81 | 309.26 | 20.74 | **290.40** | **37.19** |

*Table 7.* **Prefill and decode latency comparison under long contexts.** For each method, "Pre." denotes prefill latency, measured as time-to-first-token (TTFT) in seconds, and "Dec." denotes decode latency in milliseconds per token at batch size 1. OOM denotes out-of-memory.

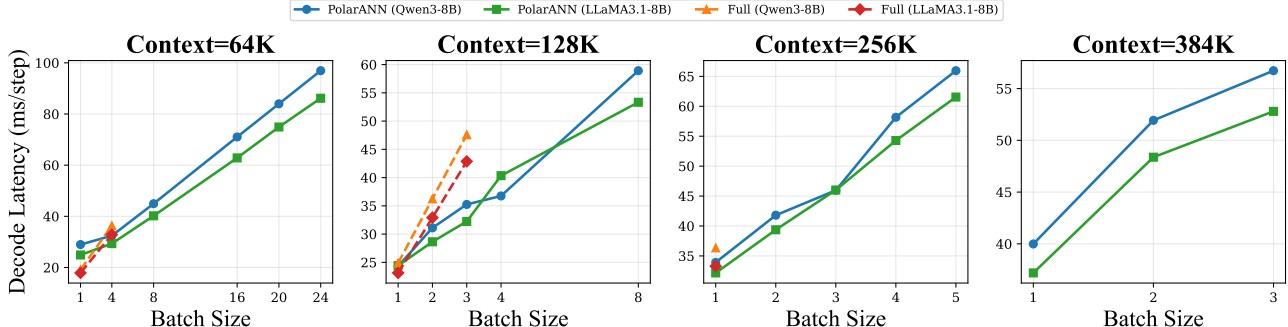

*Figure 11.* Time per output token (TPOT) vs. Context Length and Batch Size (qwen3-8b vs llama3.1-8b)

**Decode Latency (TPOT).** Fig. 11 reports the decoding TPOT (ms/step) as a function of batch size under different context lengths. Overall, ParisKV scales smoothly with batch size: TPOT increases monotonically, while the normalized per-token latency (ms/step ÷ bs) decreases due to amortization.

At 128K, ParisKV is already comparable to full attention at bs=1 (e.g., $24.32$ms/step vs. $24.87$ms/step on Qwen3-8B), and remains stable as batch size increases ($24.32 \rightarrow 58.92$ ms/step from bs=1 to bs=8). In contrast, full attention becomes memory-bounded and hits OOM at bs$\geq$4.

The gap further widens at longer contexts. At 256K, full attention only runs at bs=1 and OOMs at bs$\geq$2, while ParisKV continues scaling up to bs=5 with TPOT around 60–66 ms/step across both models. At 384K, full attention OOMs even at bs=1, whereas ParisKV remains runnable and scales to bs=3. These results explain ParisKV's consistently higher peak decoding throughput in Fig. 7.

**End-to-End Latency across KV-cache Retrieval Systems** We further compare the end-to-end decoding latency of MagicPIG, PQCache, and ParisKV under longer contexts ranging from 128K to 1024K (Fig. 7, bs=1). The results show that ParisKV's decoding latency increases only moderately as the context length grows, whereas both PQCache and MagicPIG exhibit significantly higher decoding latency at long contexts—especially PQCache, whose overhead rises sharply with context length.

Taking Llama3.1-8B as an example, ParisKV achieves 10.0× / 23.5× / 32.5× / 38.9× / 44.4× lower decode latency than PQCache at 128K / 256K / 384K / 512K / 1024K, respectively. ParisKV also consistently outperforms MagicPIG under the same contexts, and the gap further widens as the context length increases. Compared with full attention, ParisKV remains in the same order of magnitude for decoding latency within the runnable context range; however, when the context reaches 384K and beyond, full attention runs into OOM and cannot support inference.

**Prefill Latency**. Fig. 8 compares the prefill latency across methods under increasing context lengths. Overall, ParisKV shows *comparable* prefill time to prior approaches, with differences largely attributable to additional one-time preprocessing in our pipeline (e.g., quantization, codebook construction/lookup, and KV offloading setup). Since prefill is incurred once per request while decoding cost accumulates over many generated tokens, we focus our discussion on decoding latency/throughput in the main text, where ParisKV delivers clear and consistent speedups.

