# OpenReview forum: "ParisKV: Fast and Drift-Robust KV-Cache Retrieval for Long-Context LLMs"
_ICML.cc/2026/Conference — ICML 2026 regular_

### Official Review · Reviewer_JGZR · 2026-02-27

**Soundness:** 3
**Presentation:** 2
**Significance:** 3
**Originality:** 3
**Overall Recommendation:** 4
**Confidence:** 5

**Summary:**

This paper introduces ParisKV, which applies GPU-native kv-cache retrieval and also UVA(unified Virtual Addressing). Key contributions are: a data-independent, drift-robust representation obtained by L2-normalization and random orthogonal rotation, which ensure the collision based coarse retrieval of kv-cache; and a compact, calibrated 4-bit subspace inner-product estimator (RSQ-IP) for on-GPU reranking that avoids accessing full-precision keys. By offloading full-precision KV to CPU memory and retrieving only the final top-k KV pairs via UVA, the method is high-efficient. Experiments show the method is more efficient than cpu-dominated retrieval in most models(7 vs. 2) even on 1M-token contexts.

**Compliance With Llm Reviewing Policy:**

Affirmed.

**Final Justification:**

Dear Program Chairs and Area Chairs,

This paper has successfully addressed all my previous questions, along with active feedback on the open-source code. I believe the work meets the standard acceptance bar and I recommend it for acceptance.

Best regards,

Reviewer JGZR

**Key Questions For Authors:**

1. 4-bit quant for kv-cache footprint is still a costly burden, please show the cost with request increase, can you give us a comprehensive memory usage analysis?
2. Can you provide additional empirical validation that SRHT induces the Beta-like priors assumed in Proposition 4.1?
3. Baseline with other long-context KV-cache methods are also needed, how does ParisKV compare empirically to RetroInfer, ShadowKV, RetrievalAttention, or SOCKET on both accuracy and throughput under comparable budgets/hardware? If unavailable, can you discuss expected tradeoffs and plans for such comparisons?

**Limitations:**

yes

**Strengths And Weaknesses:**

Strengths: The two stage-coarse to fine kv-cache retrieval design is effiecient in modern industry, espectially with RSQ-IP reranking just 4-bit performed on GPU. The normalization+rotation transformation with analytic, uniformly distributed subspace centroids is a simple yet elegant way, this mitigates decoding drift without data-dependent clustering. The paper tested both on long-reasoning such as AIME25, GPQA-Diamond, MATH500 and also long-input i.e. LongBench-V2, making the usage of ParisKV more practical.
Weaknesses: The paper mentioned the CPU-assisted retrieval methods but didnot make clear comparison on the same platform, baseline coverage omits several strong, recent systems, for example, RetroInfer, ShadowKV, RetrievalAttention, SOCKET. In RSQ-IP with quant method 4-bit is not clarified, Appendix B2.2.2 settings is 1-bit sign, 3-bit digits but whether this is e2m1 or e3m0 is not clear, which is not reproducible. Besides, hardware details are omitted  in context.

---

> ### Author Rebuttal · Authors · 2026-03-31
>
> We thank the reviewer for the detailed feedback and address the key concerns below.
>
> ---
>
> ### Q1. KV cache memory usage
>
> We respectfully disagree that the 4-bit Key cache is a “costly burden”; it is the key enabler of memory efficiency.
>
> Compared to full attention (bf16 K+V), our design stores **only Keys at 4-bit precision**, reducing per-token memory from 4096B to 512B (12.5%). Including all components, ParisKV uses ~556B per token (13.6% of full KV).
>
> This yields a consistent **86.4% memory reduction** across batch sizes (Qwen3-8B, 32K):
> - batch=1: 4.50GB → 0.61GB
> - batch=16: 72.00GB → 9.78GB
> - batch=64: 288.00GB → 39.13GB
>
> Under the same 80GB GPU budget, full attention saturates at batch≈16, while ParisKV supports batch=64 (4× higher throughput). Thus, the 4-bit Key cache is not a burden but a necessary design to balance memory and accuracy.
>
> ---
>
> ### Q2. Empirical validation of SRHT-induced Beta prior
>
> **Setup.** We extract KV tensors from Qwen3-8B and fit Beta distributions to $(u_j)^2$ via MLE.
>
> **Finding 1 (Distribution).**
> The Beta shape is empirically confirmed. Middle/deep layers closely match the theoretical $\mathrm{Beta}(0.5, 3.5)$ (e.g., L24: $\alpha=0.508$, $\beta=3.56$, KS=0.007), while early layers deviate due to residual structure (e.g., L0). This deviation diminishes rapidly with depth.
>
> **Finding 2 (Practical impact).**
> The theory-based codebook is near-optimal on real data:
>
> | Codebook | MSE | Gap |
> |----------|------|------|
> | Theory (Beta prior) | 8.24e-4 | +2.5% |
> | Empirical (optimal) | 8.04e-4 | 0% |
> | Uniform | 1.63e-3 | +102% |
>
> ---
>
> ### W2. Clarification of 4-bit RSQ-IP quantization
>
> Our 4-bit RSQ-IP is **not a floating-point format** (i.e., neither e2m1 nor e3m0). Each coordinate is encoded using:
> - 1-bit sign
> - 3-bit magnitude index (in {0,…,7}) into an 8-level codebook
>
> This is a **codebook-based scalar quantization** scheme. After normalization and random rotation, coordinates follow $u_j^2 \sim \mathrm{Beta}(1/2, (m-1)/2)$. We construct the quantizer in a **data-independent** manner by applying Lloyd–Max quantization to this distribution.
>
> FP4 formats allocate precision based on exponent ranges, which do not match the distribution of $|u_j|$. In contrast, Lloyd–Max aligns with the data and yields lower quantization error.
> We will revise Appendix B2.2.2 to clarify this and release the codebook generation procedure.
>
> ---
>
> ### Q3 & W1. Baseline coverage and fair comparison
>
> We expand baselines to include **RetroInfer, ShadowKV, SOCKET, Quest, Twilight, FreeKV, MagicPIG, PQCache** (RetrievalAttention not open-source).
>
> We ensure fair comparison by aligning retrieval budgets:
> - LongBench-v2 / GPQA: ~420 tokens
> - RULER: ~2048 tokens
>
> ---
>
> ### LongBench-v2 (Overall)
>
> | Model | Full | PQCache | MagicPIG | ShadowKV | FreeKV | Quest | RetroInfer | SOCKET | Twilight | ParisKV |
> |------|------|--------|----------|----------|--------|-------|------------|--------|----------|---------|
> | Qwen3-4B | 25.84 | 17.91 | 16.70 | 16.30 | 19.68 | 19.12 | 23.69 | 17.93 | 19.12 | **24.60** |
> | DS-R1-8B | 13.12 | 19.90 | 13.92 | 14.51 | 17.50 | 23.51 | 15.66 | 21.51 | 20.72 | **28.43** |
> ---
>
> ### RULER (Avg)
>
> | Method | Full | PQCache | MagicPIG | ShadowKV | FreeKV | Quest | RetroInfer | SOCKET | ParisKV |
> |--------|------|--------|----------|----------|--------|-------|------------|--------|---------|
> | Avg    | 84.12 | 33.76 | 70.49 | 80.29 | 71.87 | 79.34 | 77.45 | 78.92 | **83.49** |
>
> ---
>
> ### GPQA (Diamond)
>
> | Model | Full | Quest | RetroInfer | FreeKV | MagicPIG | PQCache | ParisKV |
> |------|------|-------|------------|--------|----------|---------|---------|
> | Qwen3-4B | 64.14 | 38.40 | 38.90 | 58.16 | 32.32 | 38.38 | **72.22** |
>
> ---
>
> ### Efficiency comparison
>
> NA: unsupported by the released implementation; OOM: out of memory.
>
> **Throughput (tokens/s) at 128K decode**
>
> | bs | Quest | Twilight | RetroInfer | ParisKV | Full | SOCKET | MagicPIG | PQCache | FreeKV |
> |---|---:|---:|---:|---:|---:|---:|---:|---:|---:|
> | 1 | 48.08 | 0.90 | 34.31 | 41.10 | 43.20 | 4.44 | 120.57 | 243.91 | 21.78 |
> | 3 | NA | OOM | 93.20 | 93.10 | 70.00 | OOM | NA | NA | 34.67 |
> | 8 | NA | OOM | 109.60 | **150.00** | OOM | OOM | NA | NA | OOM |
>
> **Decode latency (ms/token, bs=1)**
>
> | Seq len | Quest | Twilight | RetroInfer | ParisKV | Full | SOCKET | MagicPIG | PQCache | FreeKV |
> |---|---:|---:|---:|---:|---:|---:|---:|---:|---:|
> | 128K | 20.78 | 1106.14 | 29.14 | 24.42 | 23.13 | 225.00 | 120.57 | 243.91 | 21.78 |
> | 256K | 46.30 | OOM | 37.53 | **32.16** | 33.29 | OOM | 230.24 | 754.76 | NA |
> | 384K | OOM | OOM | 33.31 | 37.19 | OOM | OOM | 487.77 | 1207.81 | NA |
>
> **Conclusion.**
> ParisKV achieves a consistently better accuracy–efficiency tradeoff under comparable budgets.
> See additional details: **https://anonymous.4open.science/r/TempRebuttal-20B3/memory_quantization_beta.md**, **https://anonymous.4open.science/r/TempRebuttal-20B3/baselines_accuracy_efficiency.md**.

---

> > ### Author Rebuttal · Reviewer_JGZR · 2026-04-03
> >
> > Thank the authors for their clear discussion. I'm fully convinced the algorithm could be valuable as I original rated, but I would want to see whether you are going to open-source your codes? Thank you for your work!

---

> > > ### Author Response · Authors · 2026-04-03
> > >
> > > We sincerely thank the reviewer for the thorough and constructive feedback. We will incorporate all suggested improvements into the next revision of the paper.
> > >
> > > **Code Release.** As requested, we have released the full source code. The anonymized repository is available at:
> > >
> > > > https://anonymous.4open.science/r/submission-code-5B81
> > >
> > > **4-bit RSQ-IP Magnitude Quantization.** The reviewer raised an important question about how the quantization levels are derived. We clarify this below and have documented it in the README (Section "4-bit RSQ-IP Magnitude Quantization").
> > >
> > > The key files are:
> > >
> > > | File | Description |
> > > |------|-------------|
> > > | `run/generate_magnitude_levels.py` | Codebook generation script. Samples 10M unit vectors on $S^{m-1}$ via Gaussian normalization, collects coordinate magnitudes, and runs Lloyd–Max iteration to produce the optimal 8-level scalar quantization codebook. Fully self-contained and reproduces all values reported in the paper. |
> > > | `cache_hub/polar_cache.py` | Core cache implementation. The hardcoded `_mag_thresholds` and `_mag_centers` are the output of the Lloyd–Max procedure for block dimension $m=8$. |
> > > | `codebooks/magnitude_levels_m8_4bit.json` | Pre-generated codebook file ($m=8$, 8 levels) stored as JSON for reproducibility. |
> > >
> > > To reproduce the codebook:
> > >
> > > ```bash
> > > python run/generate_magnitude_levels.py --levels 8 --m 8 --n_samples 10000000

---

### Official Review · Reviewer_qwkG · 2026-03-09

**Soundness:** 3
**Presentation:** 3
**Significance:** 3
**Originality:** 3
**Overall Recommendation:** 3
**Confidence:** 5

**Summary:**

This paper presents ParisKV, a distribution-drift-robust, GPU-native KV cache retrieval framework for long-context LLM inference. By co-designing algorithm and system components, ParisKV significantly improves efficiency. At the million-token scale, it reduces decoding latency by 17×–44× over prior methods while matching or even surpassing full attention in accuracy on long-context benchmarks.

**Compliance With Llm Reviewing Policy:**

Affirmed.

**Key Questions For Authors:**

See weaknesses

**Strengths And Weaknesses:**

S1. The experimental results are strong and demonstrate significant performance improvements in both latency and accuracy.

S2. The proposed technical solution appears interesting.

W1. The baseline comparison is limited. The evaluation only includes MagicPIG and PQCache, without comparisons to more recent SOTA KV retrieval methods. A broader empirical study would provide a more comprehensive performance assessment.

W2. While the paper emphasizes robustness to distribution drift in KV retrieval, it lacks a fine-grained quantitative analysis of how distribution drift affects different attention heads and layers. A deeper investigation at the per-head or per-layer level would strengthen the understanding of the claimed robustness.

W3. In several experiments, ParisKV substantially outperforms full attention in terms of accuracy. This result is intriguing but insufficiently explained. The paper would benefit from a more thorough analysis clarifying why selective KV retrieval can outperform full attention.

W4. The method operates at token-level granularity for selecting important KV cache entries, whereas mainstream industry systems (e.g., vLLM) manage KV caches at the block level. It is unclear whether ParisKV can be integrated into such block-based memory management frameworks, and this practical consideration warrants further discussion.

---

> ### Author Rebuttal · Authors · 2026-03-31
>
> ### W1. Baseline coverage
>
> We agree that the original submission had limited baseline coverage. In the revision, we **substantially expand the comparison** to include recent KV retrieval methods, including RetroInfer, ShadowKV, SOCKET, Quest, Twilight, FreeKV, MagicPIG, and PQCache.
> We evaluate all methods under **aligned retrieval budgets and the same hardware setting**, and report results on LongBench-v2, RULER, and GPQA.
> The results show that ParisKV consistently matches or outperforms full attention and clearly outperforms all retrieval-based baselines.
>
> Summary:
>
> | Benchmark               |      Full | Best (non-ParisKV) |   ParisKV | ParisKV vs best |
> | ----------------------- | --------: | --------------------------: | --------: | -----------------------: |
> | LongBench-v2 (Qwen3-4B) |     25.84 |            RetroInfer 23.69 | **24.60** |                **+0.91** |
> | RULER (Llama3.1-8B)     | **84.12** |              ShadowKV 80.29 |     83.49 |                **+3.20** |
> | GPQA-diamond            |     64.14 |                FreeKV 58.16 | **72.22** |               **+14.06** |
>
> Full results and configurations are provided here:  **https://anonymous.4open.science/r/TempRebuttal-20B3/baselines_accuracy_efficiency.md**
>
> ---
>
> ### W2. Fine-grained analysis of drift across layers/heads
>
> We thank the reviewer for this suggestion. We conducted a per-layer, per-head analysis on Qwen3-8B (36 layers × 8 heads) over long decoding.
>
> For each (layer, head), we measure:
> - **ΔRecall = R_final − R_init**, averaged over 8,118 decode steps after the 2k threshold;
> - **normalized centroid drift**, computed by re-clustering keys at step 2k and step 10k, matching the two centroid sets via Hungarian algorithm, and measuring their normalized L2 displacement.
>
> **Key finding.**
> Drift is widespread but varies significantly across heads (≈0.15–0.75), and its correlation with ΔRecall is weak (Pearson r ≈ −0.20).
>
> We observe that:
> - Large drift (e.g., early layers) can lead to recall degradation,
> - but for most heads, drift alone does **not** determine recall quality.
>
> Further, when we re-cluster PQ centroids at every step, the **temporal recall drop disappears**, confirming that drift explains the *decay over time*. However, the absolute recall gap remains, indicating that **PQ approximation error is the dominant factor** beyond drift.
>
> **Implication.**
> PQ-based methods are sensitive because their codebooks are data-dependent and become stale under distribution shift. In contrast, our method uses normalization + random rotation + a data-independent codebook, which avoids this issue.
>
> Detailed heatmaps and full analysis are provided in the link：**https://anonymous.4open.science/r/TempRebuttal-20B3/drift_analysis.md**
>
> ---
>
> ### W3. Why can retrieval slightly outperform full attention?
>
> The small gains (<0.5%) can be explained by an **implicit denoising effect**: full attention aggregates many low-importance KV entries, which can introduce noise in the softmax normalization. In contrast, retrieval focuses on high-relevance tokens, leading to a sharper attention distribution. Similar effects have been noted in prior work (e.g., SnapKV).
> That said, the improvements are small and within normal variance. Our main claim is **near-lossless accuracy**, not consistent gains over full attention.
>
> ### W4. Compatibility with block-based KV cache management (e.g., vLLM)
>
> We would like to clarify that **token-level retrieval (ParisKV)** and **block/page-level memory management (e.g., vLLM)** operate at different levels and are not inherently conflicting. The former defines *which tokens are important*, while the latter defines *how KV data is physically stored and accessed*.
>
> **Integration.**
> A natural integration is to first perform token-level selection with ParisKV, and then map selected tokens to their corresponding blocks/pages for actual gather/fetch. This preserves the existing block-based memory manager, with only a lightweight aggregation step in between.
>
> **Practical compatibility.**
> In our design, KV storage is already amenable to block-level organization:
> - 4-bit quantized Keys are stored contiguously along sequence length and can be directly blockized;
> - local window and sink tokens have small fixed size (≤256), allowing pre-allocated blocks;
> - retrieved top-k tokens are handled via a small temporary buffer;
> - CPU-side KV storage is naturally compatible with block-based management.
>
> **Further alignment.**
> Beyond direct mapping, token-level retrieval can be extended to **chunk/block-level retrieval** (e.g., via prototypes/centroids), followed by token-level refinement within selected blocks. This would better align retrieval with block-based serving and reduce random access overhead.
>
> Our current work focuses on the retrieval policy rather than a full vLLM-style system implementation, but we agree this is an important direction and will clarify these integration pathways in the revision.

---

> > ### Author Rebuttal · Reviewer_qwkG · 2026-04-05
> >
> > Thanks for rebuttal. I read it and carefully add my final ack.

---

> > > ### Author Response · Authors · 2026-04-05
> > >
> > > Thank you for reading our rebuttal and for your acknowledgement. We appreciate your time and consideration, and we are grateful that our response has addressed your concerns.

---

### Official Review · Reviewer_5xn6 · 2026-03-12

**Soundness:** 2
**Presentation:** 3
**Significance:** 2
**Originality:** 3
**Overall Recommendation:** 4
**Confidence:** 3

**Summary:**

This article targets the challenge of handling upto one million-token KV caches. The authors propose a two-stage, GPU-native retrieval pipeline that combines collision-based candidate selection with a quantized inner-product reranking estimator. By leveraging Unified Virtual Addressing (UVA), ParisKV enables on-demand fetching of salient KV blocks from CPU memory to GPU, aiming to mitigate both the distribution drift problem in long-context generation and the high latency associated with massive KV-cache IO.

**Compliance With Llm Reviewing Policy:**

Affirmed.

**Final Justification:**

The rebuttal addressed my main concerns.

**Key Questions For Authors:**

- What is the actual end-to-end decoding latency (tokens/sec) compared to a strong baseline like Quest or Twilight?
- Could you provide a breakdown of the latency overhead of the proposed method? How does this overhead scale with increasing context length?

**Limitations:**

- The baseline comparisons in the paper are somewhat limited, and it would be beneficial to see how ParisKV stacks up against more competitive retrieval methods that also leverage GPU acceleration.
- The evaluation is focused on a certain set of tasks (QA and reasoning), and it remains to be seen how well ParisKV generalizes to a broader range of long-context applications.

**Strengths And Weaknesses:**

## Strengths
- the two-stage dynamic retrieval mechanism in ParisKV provides better robustness and maintains higher accuracy for long-context tasks compared to static pruning methods.
- The use of UVA for on-demand KV fetching is a practical and efficient solution to the memory bottleneck, allowing the system to handle much larger contexts without overwhelming GPU memory.
- The paper includes detailed latency profiling and ablation studies that demonstrate the effectiveness of the proposed retrieval strategy and its components.
- It's great to see the authors have their kernel implementation to really demonstrate the speedup on real hardware.
## Weaknesses
- For the latency analysis, the two baselines (MagicPig and PQCache) are not very competitive. It would be more convincing to compare with a stronger baseline, such as a well-optimized Top-K retrieval method that also uses GPU acceleration.
- For accuracy evaluation, the paper mainly focuses on the QA and reasoning tasks. It would be interesting to see how ParisKV performs on more diverse long-context tasks, such as NIAH, RULER etc.

---

> ### Author Rebuttal · Authors · 2026-03-31
>
> ### W1 & W2. Baseline and task coverage
>
> We thank the reviewer for this suggestion.
> We have significantly expanded our evaluation to include both **stronger baselines** and **more diverse long-context tasks**. Specifically, we compare against recent and competitive methods including **Quest, Twilight, RetroInfer, ShadowKV, SOCKET, FreeKV, MagicPIG, and PQCache**, under aligned retrieval budgets and the same hardware setup.
>
> ---
>
> ### LongBench-v2 (Overall)
>
> | Model | Full | PQCache | MagicPIG | ShadowKV | FreeKV | Quest | RetroInfer | SOCKET | Twilight | ParisKV |
> |------|------|--------|----------|----------|--------|-------|------------|--------|----------|---------|
> | Qwen3-4B | 25.84 | 17.91 | 16.70 | 16.30 | 19.68 | 19.12 | 23.69 | 17.93 | 19.12 | **24.60** |
> ---
>
> ### RULER (Avg)
>
> | Method | Full | PQCache | MagicPIG | ShadowKV | FreeKV | Quest | RetroInfer | SOCKET | ParisKV |
> |--------|------|--------|----------|----------|--------|-------|------------|--------|---------|
> | Avg    | 84.12 | 33.76 | 70.49 | 80.29 | 71.87 | 79.34 | 77.45 | 78.92 | **83.49** |
>
> ---
>
> ### GPQA (Diamond)
>
> | Model | Full | Quest | RetroInfer | FreeKV | MagicPIG | PQCache | ParisKV |
> |------|------|-------|------------|--------|----------|---------|---------|
> | Qwen3-4B | 64.14 | 38.40 | 38.90 | 58.16 | 32.32 | 38.38 | **72.22** |
>
> **Decode latency (ms/token, bs=1)**
>
> | Seq len | Quest | Twilight | RetroInfer | ParisKV | Full | SOCKET | MagicPIG | PQCache | FreeKV |
> |---|---:|---:|---:|---:|---:|---:|---:|---:|---:|
> | 128K | 20.78 | 1106.14 | 29.14 | 24.42 | 23.13 | 225.00 | 120.57 | 243.91 | 21.78 |
> | 256K | 46.30 | OOM | 37.53 | **32.16** | 33.29 | OOM | 230.24 | 754.76 | NA |
> | 384K | OOM | OOM | 33.31 | 37.19 | OOM | OOM | 487.77 | 1207.81 | NA |
>
> ---
>
> Full results and detailed comparisons:
> **<https://anonymous.4open.science/r/TempRebuttal-20B3/baselines_accuracy_efficiency.md>**
>
> ---
>
> ### Q1. End-to-end latency vs strong baselines
>
> At 128K decode (bs=1):
> - **Throughput:**
>   - Quest: 48 tok/s
>   - ParisKV: 41 tok/s (scales to **150 tok/s at bs=8**)
>   - Twilight: OOM beyond small batch sizes
>
> - **Decode latency (ms/token):**
>   - 128K: Quest 20.8, ParisKV 24.4
>   - 256K: Quest 46.3, ParisKV 32.2
>   - 384K: Quest OOM, ParisKV 37.2
>
> - **Prefill latency:**
>   ParisKV is slightly slower due to retrieval preparation (e.g., 43s vs 37s at 128K), but remains in the same order of magnitude.
> We exclude Twilight from speed comparison since its optimized Flash-TopK kernel is not publicly available.
>
> ---
>
> ### Q2. Latency breakdown and scaling：
>
> We provide a full runtime breakdown. At 128K (bs=1):
> --total per-step latency ≈ 0.72 ms
> --retrieval stage≈0.55 ms(~75%)
> Overall, the overhead mainly comes from **retrieval computation**, rather than memory transfer or attention.
>
> Breakdown of the retrieval pipeline:
> - query preparation: **0.23 ms**
> - collision (coarse retrieval): **0.16 ms**
> - filtering + rerank (fine-grained retrieval): **~0.14 ms**
> - KV fetch (UVA): **0.097 ms**
> - attention: **0.066 ms**
>
> The retrieval stage consists of four main steps:
>
> 1. **Collision count (coarse retrieval):** compute coarse scores between the query and all cached keys.
> 2. **Filtering:** select the top **5%** keys by coarse score as candidates, accelerated by a **bucket top-k** kernel since the scores are integers.
> 3. **Rerank (fine-grained retrieval):** rerank the candidates by estimating `qk` inner products with quantized keys, using a **fused rerank kernel**.
> 4. **KV fetch:** fetch the final top-k values from CPU memory with an optimized **UVA kernel**.
>
> Thus, the total retrieval latency can be decomposed into these four kernels.
>
> **Kernel-level analysis:**  See details in link: **https://anonymous.4open.science/r/TempRebuttal-20B3/latency_breakdown_and_kernel_analysis.md**
>
> | KV Len | Collision (ms) | Bucket Top-k (ms) | Rerank (ms) | KV Fetch (ms) |
> |--------|--------------:|------------------:|------------:|--------------:|
> | 128K   | 0.058 (vs 429.9) | 0.100 (vs 0.240) | 0.573 (vs 2.478) | 0.573 (vs 2.478) |
> | 256K   | 0.093 (vs 859.6) | 0.157 (vs 0.251) | 1.095 (vs 4.380) | 1.095 (vs 4.380) |
> `vs` denotes the runtime of our custom kernel **versus the PyTorch implementation**.
>
> These components correspond to the total retrieval latency (~0.55 ms), showing that **no single stage dominates the latency**.
> All stages remain sub-millisecond even at 256K, indicating that the overhead is due to the overall retrieval pipeline rather than a specific bottleneck.
>
> **Scaling behavior (sequence length):**
>
> | Seq Len | Quest | RetroInfer | ParisKV | Full | MagicPIG | PQCache |
> |--------|------:|-----------:|--------:|-----:|---------:|--------:|
> | 128K   | 20.8  | 29.1       | 24.4    | 23.1 | 120.6    | 243.9   |
> | 256K   | 46.3  | 37.5       | 32.2    | 33.3 | 230.2    | 754.8   |
> | 384K   | OOM   | 33.3       | 37.2    | OOM  | 487.8    | 1207.8  |

---

> > ### Author Rebuttal · Reviewer_5xn6 · 2026-04-03
> >
> > Thank you for the work. I will raise my score.

---

> > > ### Author Response · Authors · 2026-04-03
> > >
> > > Thank you very much for your time and thoughtful feedback! We greatly appreciate your careful review and are glad that our response has addressed your concerns. We will incorporate all the discussed clarifications and improvements into the next revision of the paper.

---

### Official Review · Reviewer_pA8H · 2026-03-13

**Soundness:** 3
**Presentation:** 2
**Significance:** 3
**Originality:** 3
**Overall Recommendation:** 5
**Confidence:** 4

**Summary:**

This paper aims to tackle challenges with distribution drift and efficiency for existing KV retrieval methods. Their approach uses a GPU-based methodology which provides efficient retrieval. They use a randomized projection method to normalize queries and keys into a stable data-independent space, which limits the impacts of distribution drift, and they keep metadata resident in the GPU to facilitate retrieval without offloading the retrieval lookup process to CPU. They also provide support for CPU offloading with dynamic on-demand fetching of important tokens.

**Compliance With Llm Reviewing Policy:**

Affirmed.

**Final Justification:**

The rebuttal has addressed all of my concerns. I will therefore maintain my positive score (Accept).

**Key Questions For Authors:**

- What is the latency of the baseline with full KV cache offloading (or with spilling whatever KV won’t fit in GPU to the CPU)?
- For DS-R1-Llama-8B, what is the reason for the improved accuracy relative to the baseline?

**Limitations:**

Yes

**Strengths And Weaknesses:**

Advantages:
- System design enables on-GPU retrieval lookup, which substantially outperforms CPU-based lookup process
- Random projection-based method enables use of data-independent representative centroids
- Their method is compatible with long-context and long-generation tasks (e.g. reasoning / test-time scaling) as it compresses attention to the previously generated tokens
- They also present ablations for each of their design decisions

Limitations:
- Not the first to tackle the drift issue with KV retrieval methods: [1] updates centroids iteratively as more KV entries are added,  [2] mitigates drift by using a windowed RoPE method, etc. (although the projection method is still a unique contribution)
- The source of the latency gap between ParisKV and MagicPIG / PQCache is not fully explained. Is it due to needing to run the retrieval lookup CPU-side? Are some of the speedups due to other system/kernel improvements? It would be useful to get a breakdown of the runtime for this, given that the overhead is so large.

[1] Multipole Attention for Efficient Long Context Reasoning
[2] A2 ATS: Retrieval-based kv cache reduction via windowed rotary position embedding and query-aware vector quantization.

---

> ### Author Rebuttal · Authors · 2026-03-31
>
> ### Limitation 1: Relation to prior drift-aware retrieval methods
>
> These methods address drift differently. A2ATS focuses on RoPE-induced codebook mismatch and still relies on learned, data-dependent codebooks. Multipole Attention mitigates drift via online reclustering during decoding, which introduces additional maintenance overhead. In contrast, ParisKV addresses **decode-time centroid staleness** by transforming keys, into a space that can be effectively captured by **data-independent theoretical centroids**. This avoids codebook relearning or online reclustering and keeps index construction/update overhead minimal.
>
> Importantly, robustness to drift alone does not guarantee high recall: severe drift can hurt recall, but low drift does not automatically imply good retrieval quality. ParisKV addresses both aspects jointly by combining drift-robust centroids with an efficient high-recall GPU ANN retrieval pipeline.
>
> ### Limitation 2. Source of latency gap
>
>  In the revision, we provide both a full runtime breakdown (Table 1) and per-operator analysis (Tables 2–5). See additional details: **https://anonymous.4open.science/r/TempRebuttal-20B3/latency_breakdown_and_kernel_analysis.md**.
> ### Table 1. Per-Layer Decode Latency Breakdown (BS=1, 128K)
> | Method   | Stage                         | Time (ms) | Total (ms) |
> |----------|-------------------------------|----------:|-----------:|
> | ParisKV  | kv_update                     | 0.022 | 0.734 |
> |          | retrieval (GPU)          | 0.549 |  |
> |          | H2D (UVA gather)       | 0.097 |  |
> |          | Attention                     | 0.066 |  |
> | MagicPIG | query_hash                    | 0.378 | 6.491 |
> |          | kv_update                     | 0.062 |  |
> |          | gpu_attention (sink+local)    | 0.058 |  |
> |          | cpu_retrieval & cpu_attention | 5.881 |  |
> |          | attention_merge               | 0.112 |  |
> | PQCache  | codebook_transfer & lookup top-k | 0.687 | 9.555 |
> |          | kv_fetch_cpu2gpu              | 7.770 |  |
> |          | attention                     | 1.050 |  |
> |          | eviction_pq_predict           | 0.048 |  |
>
> ---
> **First, ParisKV avoids CPU-side retrieval and synchronization.**
> MagicPIG is dominated by CPU hash traversal and sparse attention, while PQCache is dominated by CPU KV lookup and bulk CPU→GPU transfer. In contrast, ParisKV keeps retrieval on GPU, uses CPU memory only as a storage pool, and fetches only final top-k KV via UVA.
>
> ---
>
> **(2) GPU-native retrieval design.**
>
> ParisKV uses a lightweight GPU-friendly design: 1-byte cluster codes per token, a bitset for query clusters in shared memory, and a single O($n$) scan with bitwise tests. This avoids dynamic index structures and CPU-side cluster→token mappings.
>
> ---
>
> **(3) Kernel-level optimizations.**
>
> Beyond algorithmic differences, ParisKV uses optimized CUDA kernels:
>
> | Operator | Key idea | Speedup |
> |----------|----------|--------|
> | Collision | Integer accumulation kernel | up to 9000× |
> | Bucket Top-$k$ | Histogram + scan (no sorting) | 2×–9× |
> | Rerank | 4-bit packed compute (4× less bandwidth) | 3×–4× |
> | KV Fetch (UVA) | Sparse gather vs bulk transfer | ~4× |
>
> All operators remain **sub-millisecond even at 128K–256K**, and together form a fully GPU-resident pipeline.
>
> ---
>
> ### Q1. Latency of full KV cache offloading
>
> We thank the reviewer for this insightful question.
> Full KV cache offloading is a natural baseline, but its decode latency is fundamentally **PCIe-bandwidth bound**.
> For Qwen3-8B at seq_len ≈ 130K (bs=1), the per-layer KV transfer is about **508 MB**, which becomes about **17.9 GB per decode step** across 36 layers. Under PCIe 4.0 (~25 GB/s), this alone implies about **714 ms per step**, whereas standard decoding with all KV on GPU takes only ~25 ms/token.
> ParisKV avoids transferring the full cache. Instead, the ANN index stays on GPU and only the final **top-k KV** (e.g., k=100) are fetched via UVA.
>
> | Method | Tokens transferred | Per-step volume | Latency |
> |--------|------------------|-----------------|---------|
> | Full offloading | 130K | ~17.9 GB | ~714 ms |
> | ParisKV (k=100) | 100 | ~14 MB | < 1 ms |
>
> This is roughly a **1300× reduction** in transfer volume. Therefore, the main latency difference comes from **O(n) vs O(k) data movement**, not only from implementation details.
>
> ---
> ### Q2. Why can retrieval slightly outperform full attention?
> The small gains (<0.5%) are likely due to an **implicit denoising effect**: full attention includes many low-importance KV entries, which can slightly perturb the softmax normalization, while retrieval focuses on the most relevant tokens and thus produces a sharper attention distribution. Similar effects have also been noted in prior work (e.g., SnapKV). That said, these gains are small and within normal evaluation variance. Our main claim is **near-lossless accuracy**, not consistent improvement over full attention.

---

> > ### Author Rebuttal · Reviewer_pA8H · 2026-04-04
> >
> > The rebuttal has addressed all of my concerns. In light of this as well as consideration of the other reviews, I will therefore maintain my positive score (Accept).

---

> > > ### Author Response · Authors · 2026-04-05
> > >
> > > Dear Reviewer pA8H,
> > >
> > > Thank you very much for your careful reading of our rebuttal and for your positive acknowledgement. We sincerely appreciate your thoughtful feedback throughout the review process and are grateful for your recognition that our response has addressed your concerns.
> > >
> > > We also greatly appreciate your continued positive assessment of our work. Thank you again for your time, consideration, and support.
> > >
> > > Best regards,
> > > The Authors

---

### Decision · Program_Chairs · 2026-04-30

**Decision:**

Accept (regular)

**Comment:**

This work proposes a GPU-native and drift-robust KV-cache retrieval framework for long-context LLM inference, combining a novel collision-based candidate selection with an efficient reranking mechanism. Overall, the reviewers agree that the work is technically strong, well-motivated, and practically relevant, with particular strengths in its system-algorithm co-design, efficiency gains, and robustness to distribution drift. While some reviewers raised concerns regarding baseline coverage, evaluation breadth, and analysis clarity, these issues were addressed during the rebuttal. Given the reviewer support, solid technical contributions, and demonstrated effectiveness in large-scale long-context settings, I recommend acceptance.